# Tamm-cavity terahertz detector

Xuecou Tu [1,2] ✉, Yichen Zhang[1], Shuyu Zhou[1], Wenjing Tang[1], Xu Yan[1], Yunjie Rui[1], Wohu Wang[1], Bingnan Yan[1], Chen Zhang[1], Ziyao Ye[1], Hongkai Shi[1], Runfeng Su[1], Chao Wan[3], Daxing Dong[4], Ruiying Xu[5], Qing-Yuan Zhao [1,3], La-Bao Zhang [1,2], Xiao-Qing Jia [1,2], Huabing Wang [1,3], Lin Kang[1,2] ✉, Jian Chen [1,3] & Peiheng Wu [1,2] ✉

Efficiently fabricating a cavity that can achieve strong interactions between terahertz waves and matter would allow researchers to exploit the intrinsic properties due to the long wavelength in the terahertz waveband. Here we show a terahertz detector embedded in a Tamm cavity with a record Q value of 1017 and a bandwidth of only 469 MHz for direct detection. The Tamm-cavity detector is formed by embedding a substrate with an $Nb_5N_6$ microbolometer detector between an Si/air distributed Bragg reflector (DBR) and a metal reflector. The resonant frequency can be controlled by adjusting the thickness of the substrate layer. The detector and DBR are fabricated separately, and a large pixel-array detector can be realized by a very simple assembly process. This versatile cavity structure can be used as a platform for preparing high-performance terahertz devices and opening up the study of the strong interactions between terahertz waves and matter.

In recent years, thanks to the development of terahertz sources[1–4], detectors[5–10], modulators[11–13], and other devices[14,15], many remarkable results in imaging[16–19], molecular gas detection[20,21], and communication[22–25] in terahertz science and technology[26] have been achieved. One of the most important scientific issues for the development of these devices is how to enhance the strong interactions between these devices and terahertz signals to achieve efficient coupled input or output[27–34]. Optical resonators and nanocavities, such as Fabry–Pérot (FP) interferometers[35,36], microcavities[37–40], photonic crystals[41,42], and planar resonators[43,44], are powerful tools for producing strong interactions[45]. In particular, enhanced structures based on a Tamm cavity[46–50] with a built-in distributed Bragg reflector (DBR) are commonly used in photodetectors[51–54] and lasers[55–57]. Since Tamm cavities in the optical band based on a metal DBR were first proposed by ref. 47, they have had an important role in enhancing the interaction between material and light to realize high Q and tunable devices[58–60]. These excellent properties are necessary for terahertz-band devices. However, the functional components integrated with the DBR in the terahertz

spectral range have rarely been reported in the literature. The main difficulty is that the smallest planar features are of the order of $\lambda/4n_r \approx 10\,\mu m$, where $n_r$ is the refractive index of the dielectric. Terahertz wavelengths are in the range 10 to 1000 μm, so depositing thin films of an optical dielectric, which is commonly used for optical devices, cannot be used to construct microcavities, which is necessary for the DBR structures used in terahertz devices.

Note that the features of terahertz microcavities are of the order of the thickness of the substrates of the terahertz devices, so researchers have also begun to use the substrates as FP cavities when constructing electromagnetic confinement devices[61–63]. To obtain a tunable terahertz device, the length of the cavity can be changed with an electronically controlled displacement platform or a microelectromechanical system (MEMS) with micrometer precision[64,65]. These FP cavities have greatly facilitated the development of terahertz components that are continuously frequency adjustable and have a wideband response[66,67]. Obviously, the performance of these devices can be further improved if the Tamm cavity is used as in optical wavebands[68,69]. An optical DBR cavity composed of multiple layers of Si

[1]Research Institute of Superconductor Electronics (RISE), School of Electronic Science and Engineering, Nanjing University, Nanjing, Jiangsu 210023, China. [2]Hefei National Laboratory, Hefei 230088, China. [3]Purple Mountain Laboratories, Nanjing, Jiangsu 211111, China. [4]Department of Applied Physics, Nanjing University of Aeronautics and Astronautics, Nanjing 210016, China. [5]Nanjing Electronic Devices Institute, Nanjing 210016, China. ✉e-mail: tuxuecou@nju.edu.cn; kanglin@nju.edu.cn; phwu@nju.edu.cn

and air was fabricated by an ingenious and complex process. It has a very high refractive index contrast and very high $Q$ value. This device has been used in various high-performance lasers[70–72]. Obviously, the difficulties in depositing dielectric thin films and lateral etching are hard to address at the micro-scale in the terahertz band. Therefore, a detector or source integrated with a Tamm cavity in the terahertz band has not been reported experimentally.

Here we propose a terahertz detector embedded in a Tamm cavity, which consists of a DBR with silicon/air layers, a microbolometer detector deposited on silicon substrate, and a reflective mirror. In physics, this structure can be seen as a hybrid Tamm cavity formed by inserting a dielectric layer into a pure Tamm cavity, and the terahertz detector is prepared on this dielectric layer. The air and silicon dielectric layers are formed by deep silicon etching in the same high-resistivity float-zone (HRFZ) Si wafer chip. The silicon chip containing the air cavity is stacked and bonded with a photoresist to form a DBR with multiple Si/air layers. It is bonded to the detector chip containing a microbolometer detector and a substrate layer to form the hybrid Tamm cavity. The resonant modes of the detector can be tuned by controlling the thickness of the substrate of the microbolometer detector chip. The DBR and detector are fabricated separately, which reduces the complexity of design and fabrication. Large-scale fabrication can be achieved by simple MEMS process and bonding assembly, which is also compatible with other terahertz devices. This approach overcomes the drawbacks that millimeter-scale multilayers are hard to control precisely and integrate with terahertz devices[73].

We demonstrate experimentally that the Tamm cavity coupled detector has high $Q$ and very narrowband optical responsivity in the terahertz band. It provides a general operating platform for other devices that need enhanced interactions between matter and a terahertz wave. In particular, it can be used to study the electronics and optoelectronics of 2D materials[74–77] or to fabricate terahertz lasers, terahertz detectors, and other high-performance functional devices.

## Results
### Device design
As shown in Fig. 1a, the hybrid Tamm cavity structure is formed by sandwiching a HRFZ-Si substrate between a DBR with three Si/air layers, a terahertz detector, and a reflective metal mirror. The detector, which is a $Nb_5N_6$ microbolometer, is embedded in this structure through being deposited on the HRFZ-Si substrate that controls the electric field intensity at the detector. The detector can, in principle, be any electric field detector. It acts as a near-field terahertz probe[78]. Here, we used a $Nb_5N_6$ microbolometer whose voltage responsivity is proportional to the electric field intensity. The key to designing a hybrid Tamm cavity detector is realizing the DBR in the terahertz band and ensuring it is compatible with the detector integration process. A significant advantage of the hybrid Tamm cavity is that the detector chip and the DBR can be prepared separately, making the design and fabrication simple. The HRFZ-Si is, undoubtedly, the best choice for constructing this DBR due to the small losses in the terahertz band[79,80] and its compatibility with standard silicon MEMS processes. The air layer can be obtained by Si etching using a square opening in the same chip. More importantly, DBR made of HRFZ-Si and air has a large refractive index contrast ($n_{si} = 3.4147$ and $n_{air} = 1$), resulting in a wide transmission stopband. The thickness of the substrate of the detector chip is determined primarily according to the desired detection band, after which the thicknesses of the silicon and air layers in the DBR are optimized. To detect electromagnetic waves around 0.65 THz, according to the resonant conditions in the cavity modes, the thickness of the detector chip was 510 μm, which can support this resonant mode. According to design theory for a DBR, we let $n_H d_H = n_L d_L = \lambda/4$, where $\lambda$ is the target resonant wavelength, $n_H = 3.4147$ is the refractive index of silicon, $d_H = 33$ μm is the thickness of the silicon layer, $n_L = 1$ is

the refractive index of air, and $d_L = 115$ μm is the thickness of the air layer.

By using the electromagnetic wave transfer-matrix method (TMM) of multilayer media (Supplementary Note S1), the reflection spectrum of a DBR with three Si/air layers is calculated, as shown by the gray solid line in Fig. 1b. The DBR reflects up to 100% in the band from 0.5 to 0.8 THz. A spectrally wide stop band filter is realized just by three Si/air layers, which benefits from the high refractive index contrast between HRFZ-Si and air. The reduced period number also reduces fabrication and micro-assembly errors.

Figure 1c shows the spatial distribution of the refractive index of each layer of material of the final hybrid Tamm cavity shown in Fig. 1a. The Tamm modes[46,47] occur in this hybrid Tamm cavity under certain conditions for a one-dimensional photonic crystal. The detector substrate acts as a dielectric resonant cavity in this hybrid Tamm cavity, and the phase change is described by $4\pi n_{Si} d_{cavity}/\lambda$ in a round trip. The thickness of the substrate should satisfy the following condition if Tamm resonance occurs in the entire hybrid Tamm cavity[47,56]:

$$r_{DBR} r_{Au} e^{i(2n_{Si} d_{cavity})} = 1 \qquad (1)$$

The phase condition at the resonant point of the hybrid Tamm cavity fully conforms to the conditions for optical Tamm states (Supplementary Note S2), indicating that there is also an "optical band-like" Tamm mode in the terahertz band. This is special and different because it not only satisfies the conditions for a Tamm state but also maximizes the electric field at the position of the detector [i.e., $E_d$ in Fig. 1a]. The thickness of the detector substrate in the cavity must also satisfy the following condition:

$$d_{substrate} = \frac{(2N+1)\lambda}{4n_{Si}} \qquad (2)$$

where $N$ is the resonant mode of the FP cavity of the detector substrate. This is exactly the condition for the enhancement of the coherence of the electric field in the detector substrate layer mainly, that is, the thickness of the detector substrate determines the resonant frequency of the entire hybrid Tamm cavity. Thus, the resonant modes of this hybrid Tamm cavity are mainly determined by the thickness of substrate layer (i.e., $d_{substrate}$, the thickness of the microbolometer substrate here)[56]. This finding is verified by the following calculated results.

The blue line in Fig. 1b is the reflection coefficient of the entire hybrid Tamm-cavity. with $d_{substrate} = 510$ μm calculated by the TMM (Supplementary Note S1). Multiple resonant extremum points (Tamm modes) were formed due to the attachment of the detector chip to the substrate layer. $N$ is the resonant mode, as determined by Eq. (2). At 0.4790 THz, more than 90% of the energy is confined to the detector substrate and the $Q$ value is up to 1935. The full width at half maximum (FWHM) is only 247 MHz. The electric field distributions are calculated at the center of the hybrid Tamm cavity. [the purple dashed line in the direction of the $y$-axis in Fig. 1a] for points $A$ (0.4004 THz), $B$ (0.4300 THz), and $C$ (0.4790 THz), as shown in Fig. 1d–f. The electric field intensity at the detector $|E_d|^2$ is shown as the black dashed line. In the simulation, the electric field intensity of the incident terahertz plane wave $|E_i|$ is set as 1, and $|E|^2/|E_i|^2$ represents the enhancement factor of the electric field intensity at the purple dashed line in Fig. 1a. At point $A$, since the reflection coefficient is 0.6 and the electric field at the detector was not at a node of the standing wave, the enhancement factor was only 32. At point $B$, the electric field intensity at the detector was 0 due to total reflection of the incident terahertz wave. At point $C$, the reflection coefficient was only 0.2 and the electric field at the detector was at a node of a standing wave in the entire hybrid Tamm cavity, so the enhancement factor was a maximum of up to 57. The electromagnetic field oscillated in the substrate layer, and the energy

was confined to the substrate layer and eventually absorbed by the detector, greatly enhancing the response sensitivity of the detector. This kind of hybrid Tamm cavity significantly enhances the interaction between terahertz waves and the sensor. There is a significant difference in the electric field intensity at different locations, so it is important to precisely control the thickness of each layer of the media during device preparation. Fortunately, controlling the thickness at the micron level in deep silicon etching of MEMS is no longer a problem.

Figure 1g shows the calculated enhancement of the electric field intensity at the terahertz detector with $d_{substrate} = 510\ \mu m$ in the DBR for three cases: (1) only the substrate layer, (2) the substrate layer with one Si/air layer DBR on top, and (3) the substrate layer with three Si/air layers DBR on top. At 0.479 THz with no DBR, $|E_d|^2/|E_i|^2$ is only 3.8 and the FWHM is 15,300 MHz. When the DBR has one Si/air layer, $|E_d|^2/|E_i|^2$ increases to 24 and the FWHM is 2250 MHz. With three Si/air layers, $|E_d|^2/|E_i|^2$ is 57 and the FWHM is only 295 MHz. Notably, the resonant frequencies are consistent with the resonant frequencies of the structure with only the substrate layer. That is, the thickness of the detector chip determines the resonant frequencies of the entire cavity in the band gap of DBR, and this is consistent with the previous analysis. The corresponding reflectance of these three structures is also calculated, and to further illustrate these characteristics, we also calculated the reflection of a three-layer Si/air DBR hybrid Tamm cavity for different substrate thicknesses (Supplementary Note S3). The resonant modes shift to lower frequencies as the substrate thickness

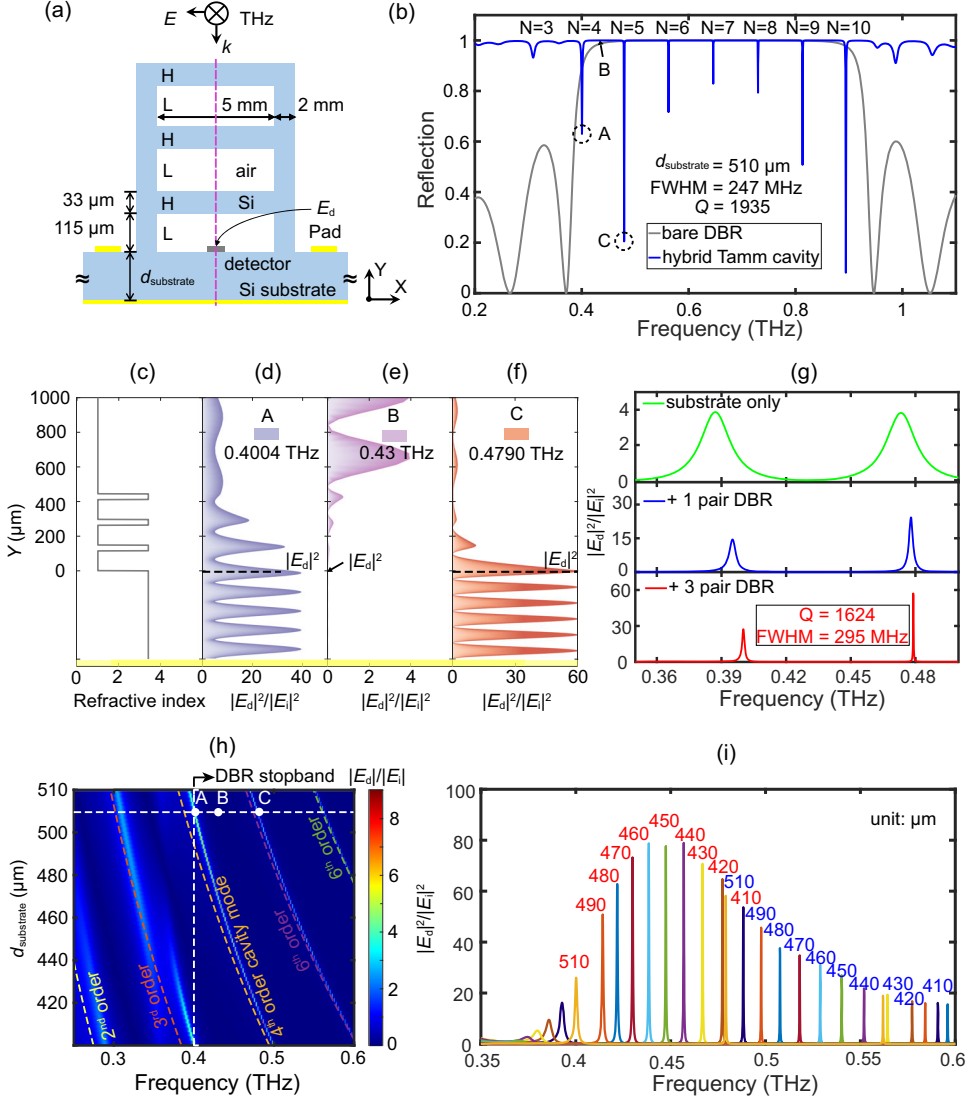

**Fig. 1 | Schematics of the hybrid Tamm-cavity detector operating in the terahertz band and its resonance characteristics. a** Schematic diagram of the hybrid Tamm-cavity detector. Silicon is shown in gray, air in white, and the mirror back on the substrate in yellow. **b** Reflectance spectra of a bare three-layer DBR and a hybrid Tamm cavity at $d_{substrate} = 510\ \mu m$. The different extremum points correspond to the cavity's resonant modes in Eq. (2). **c** Spatial distribution of the refractive index of the multilayer dielectrics in the hybrid Tamm cavity in the vertical direction. The yellow region represents the position of the metal mirror at the back of the detector chip with $d_{substrate} = 510\ \mu m$. **d–f** Spatial distributions of the enhancement factor of the electric field intensity ($|E|^2/|E_i|^2$) along the vertical purple dashed line in (**a**) at 0.4004 THz [A in (**b**)], 0.4300 THz [B in (**b**)], and 0.4790 THz [C in (**b**)]. The black dashed lines indicate the electric field intensity $|E_d|^2$ at the detector

($Y = d_{substrate} = 510\ \mu m$). **g** Electric field enhancement factor ($|E_d|^2/|E_i|^2$) at the detector with zero, one, or three Si/air layers with $d_{substrate} = 510\ \mu m$. **h** Relation between the electric field ($|E_d|$) and substrate thickness of the detector chip ($d_{substrate}$) for a hybrid Tamm cavity with three Si/air layers in the range 0.25–0.6 THz. The five white dashed lines were calculated by Eq. (2) and indicate the resonant modes of the cavity with N = 2, 3, 4, 5, or 6, respectively. The horizontal white line indicates the resonance characteristics of the hybrid Tamm cavity at $d_{substrate} = 510\ \mu m$. The dots labeled A, B, and C correspond to the cases illustrated in (**b**, **d**, **e**), and (**f**). The band gap of DBR is also marked with vertical white dashed lines on the graph. **i** Spectral characteristics of the electric field intensity $|E_d|^2$ in a hybrid Tamm cavity with different thicknesses of the substrates of the detector chips.

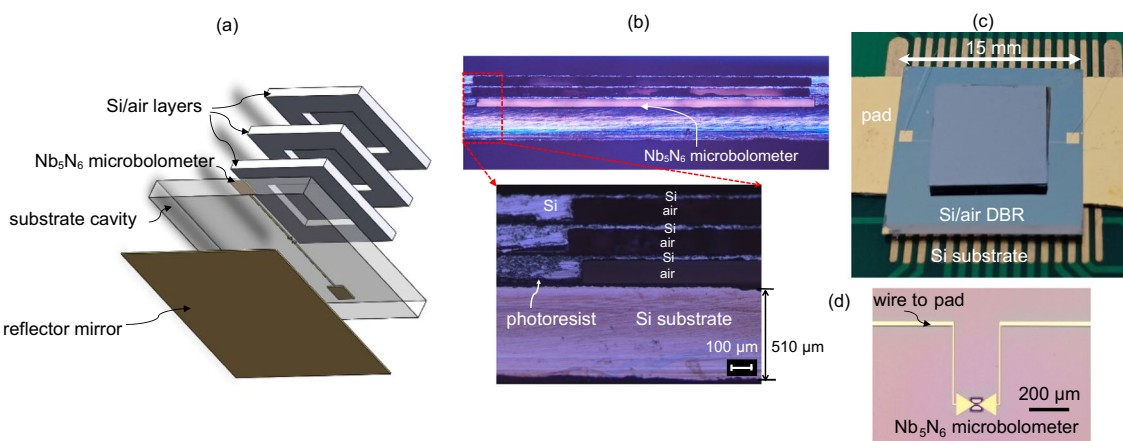

**Fig. 2 | Pictures of components of the prepared Tamm-cavity terahertz detector. a** 3D stereogram of the hybrid Tamm-cavity detector, which consists of a DBR with three Si/air layers and a Nb₅N₆ microbolometer detector. There is a metal reflector on the back of the detector chip. **b** Side view of Si/air DBR layers assembled onto the detector chip after being bonded together with a photoresist. **c** Package for a hybrid Tamm-cavity detector on a printed circuit board. **d** Nb₅N₆ microbolometer terahertz detector.

increases. This is the so-called redshift, which is consistent with the case with the substrate layer only.

The electric field at the position of the detector ($|E_d|$) is the best figure of merit. Figure 1h shows the calculated $E_d$ in the hybrid Tamm cavity for different thicknesses of the detector substrate ($d_{substrate}$) and frequencies from 0.25 to 0.60 THz. $E_d$ increases significantly at the resonance point, and the resonance is strong in accordance with Eq. (2). Prominently, there are five cavity modes, corresponding to $N = 2, 3, 4, 5,$ and 6 in Eq. (2), as indicated by the colored dashed lines. The horizontal white dashed line indicates $|E_d|$ for the substrate layer at $d_{substrate} = 510$ μm, and the white dots are $|E_d|$ at $A$, $B$, and $C$. Clearly, the resonant modes of the hybrid Tamm cavity can be adjusted by changing the substrate thickness. The resonance shifts to a higher frequency when $d_{substrate}$ is decreased, which is a blueshift, and the resonance frequencies overlapped, meaning that the low resonance mode of a thin substrate overlaps with the high resonance mode of a thick substrate, as shown in Fig. 1i. The splitting observed within the 0.35−0.4 THz region has anti-crossing effect in Fig. 1h, indicating a hybridization mode which is the strong coupling of FP cavity mode excited in detector substrate and leaky Tamm mode excited in a pure Tamm structure. As analyzed in Supplementary Note S3, this splitting is caused by the coupling between the leaky Tamm cavity mode and the detector's substrate cavity (FP mode). The leaky Tamm modes has low quality factor $Q$ and large reflectivity due to the imperfect reflection outside the DBR stopband[56,73], localizes its energy within the DBR structure, giving space for the detector substrate to excite FP cavity modes. Consequently, the hybrid mode exhibits leakiness, leading to lower electrical intensity.

When designing this kind of hybrid Tamm cavity, the target resonance points can first be calculated directly from the corresponding resonant modes of the detector chip only. Then the DBR can be designed to excite Tamm modes to couple with these FP cavity modes, enhancing the electric field at the detector without changing the resonant frequency points of the entire structure. Moreover, the resonant bandwidth can be narrowed. The detector chip and dielectric DBR are designed separately and then assembled together, which is convenient for design and fabrication. This all-silicon hybrid Tamm cavity can be used as a general platform for terahertz sources, detectors, and other functional devices. It is, possibly, the ultimate solution for achieving strong interactions between terahertz electromagnetic waves and matter.

### Device fabrication

To realize the hybrid Tamm-cavity structure with a detector embedded in, a 6-inch HRFZ-Si wafer ($\rho > 10,000$ Ω.cm) is thinned to 148 μm. An array of air cavities with a 33-μm top layer of silicon and a 115-μm bottom layer of air is formed by deep silicon etching of the same wafer. The unit size is 9 mm × 9 mm. Considering the spot size of an incident terahertz wave, the area of the opening in a silicon pixel is 5 mm × 5 mm, as shown in Fig. 2a. We cut the wafer into many single-pixel Si/air layer blocks, which are then stacked to form a multilayer DBR using a photoresist, as shown in Fig. 2a. The thickness of the photoresist distributed around the silicon support leg is about 1 μm, which has little influence on the entire hybrid Tamm cavity. For the detector chip, we used a 510-μm HRFZ-Si substrate. The Nb₅N₆ microbolometer is microfabricated through magnetron sputtering, lithography, air-bridge etching, and other micro-processing techniques. Figure 2d is an optical photograph of the finished detector chip. The DBR chip and the detector chip are micro-assembled together by a photoresist to form the hybrid Tamm cavity. Figure 2b is a side view of the hybrid Tamm-cavity detector. To read out the response voltage of the detector, the entire package is fixed to a printed circuit board [Fig. 2c]. The preparation of the hybrid Tamm-cavity detector is illustrated in detail in Supplementary Note S4.

The multilayer DBR is obtained by stacking Si/air layer blocks, which were from the same wafer, by deep silicon etching as a MEMS process. Furthermore, the detector chips and the DBR chips are prepared separately and can be assembled or disassembled. Fabricating this kind of hybrid Tamm-cavity structure is compatible with the fabrication of other terahertz functional devices, and thus, it provides an excellent platform for enhancing the interactions between terahertz waves and matter. In particular, there are many potential applications due to the strong electromagnetic coupling between the terahertz waves and the two-dimensional material.

### THz response of the Tamm-cavity detector

To verify the design of the proposed Tamm-cavity terahertz detector, three cavities coupled to a Nb₅N₆ microbolometer detector were prepared: (1) only the substrate layer (without a DBR), (2) a one-layer Si/air DBR, and (3) a three-layer Si/air DBR. Figure 3 shows the measured optical voltage responsivity of these three detectors. The measurement setup and method are described in Methods. The Si/air photonic crystal layers in this hybrid Tamm cavity significantly increases the interaction between an incident terahertz wave and the sensor, but hardly changes the resonant modes of the detector. These results are consistent with our previous simulation analysis. The findings are one of the subtleties of a hybrid Tamm cavity.

Figure 3a shows the optical voltage responsivity of the detector without a DBR. As discussed above, due to the cavity modes in the

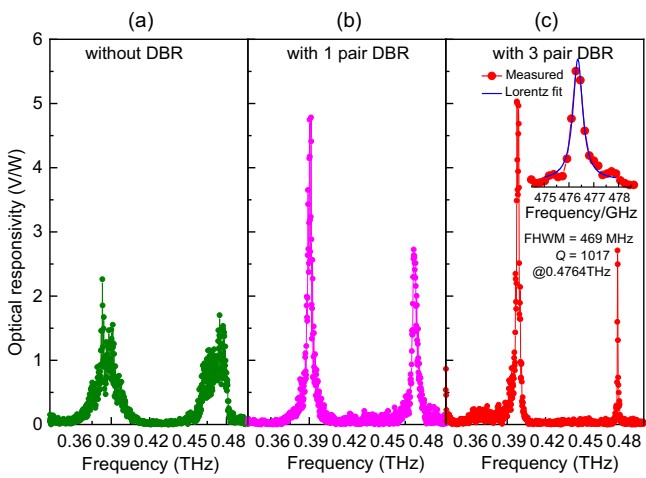

**Fig. 3 | Optical voltage responsivities of detectors. a** with a zero-layer DBR, (**b**) with a one-layer DBR, and (**c**) with a three-layer DBR. The inset is a magnified view near the resonant mode at 0.476 THz.

## Table 1 | Comparison between measured and calculated resonant frequency and FWHM

| DBR pair | Resonant frequency (THz) | | FWHM (MHz) | |
|---|---|---|---|---|
| | Cal. | Mea. | Cal. | Mea. |
| 0 | 0.473 | 0.474 | 14200 | 20609 |
| 1 | 0.478 | 0.472 | 2286 | 3901 |
| 3 | 0.479 | 0.476 | 247 | 469 |

substrate layer, the response has two resonant peaks at 0.40 and 0.48 THz. The FWHM at 0.48 THz is 20.6 GHz, and the $Q$ value is 23, as calculated by Lorentz fitting. Figure 3b shows the optical voltage responsivity when the detector has a one-layer DBR. It also resonates at about 0.40 and 0.48 THz. There is a twofold increase in the optical voltage responsivity at 0.40 THz and a 1.5-fold increase at 0.48 THz. The FWHM is 3.90 GHz, and the $Q$ value was 121 at 0.48 THz, both of which had improved by 5.3 times compared with the substrate layer only. Figure 3c shows the optical voltage responsivity when the detector has a three-layer DBR. This detector also resonates at about 0.40 and 0.48 THz. Based on Lorentz fitting, the response bandwidth and $Q$ value reached 469 MHz and 1017, respectively, as shown in the inset. To the best of our knowledge, this is the narrowest bandwidth for a terahertz detector that has been reported. Note that the optical voltage responsivity of the detector is only a little higher than that of the one-layer DBR. The escalation of dielectric loss is primarily attributed to the increasing number of DBR layers. Furthermore, deviations in the thickness of each layer and surface roughness stemming from the MEMS process notably hinder the hybrid Tamm cavity's performance. In our calculation, a mere ±3 μm fluctuation in layer thicknesses resulted in up to a 7% deviation in $Q$. Potential roughness on the silicon surface induces diffuse reflections, limiting the amount of electromagnetic wave that can be coupled to Au mirror, ultimately leading to a diminished $Q$ and electrical intensity. The voltage responses are almost zero at non-resonant frequencies, which verifies the perfect filtering characteristics of the hybrid Tamm cavity. Table 1 compares the measured and calculated $Q$ values and FWHM values at the resonant modes for the three detectors. The positions of the measured resonance peaks are almost the same as the calculated peaks. There was a slight deviation in frequency, mainly caused by errors when tuning the thickness of the DBR layers. The calculated results are for an ideal situation that neglects absorption by the layers. As shown in Fig. 4, the deviation of the $Q$ values increases as the

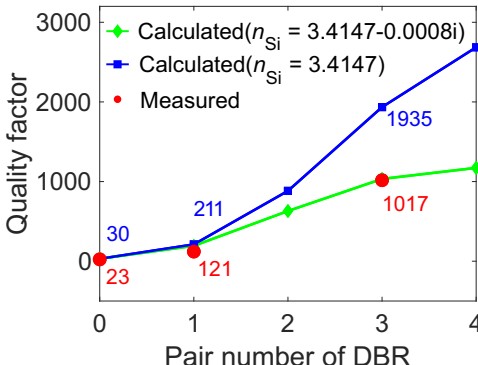

**Fig. 4 | Comparison of measured and calculated $Q$ values at a resonant mode (0.48 THz) of the detector with different pair numbers of DBR.** The calculated $Q$ values are extracted from reflection spectrum with $n_{Si} = 3.4147$ and $n_{Si} = 3.4147 - 0.0008i$, respectively.

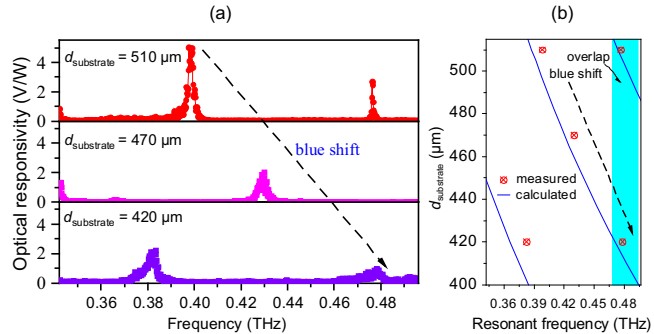

**Fig. 5 | Demonstration of the tunability of the hybrid Tamm-cavity terahertz detectors. a** Measured optical responsivity of Tamm-cavity detectors with $d_{substrate} = 510$, 470, or 420 μm. **b** Comparison of the measured and calculated resonant frequencies for $d_{substrate}$ from 510 to 420 μm. The black dashed arrow indicates the blueshift, and the cyan region is where the cavity modes overlap.

number of layers increases. The theoretical $Q$ value reaches 1935, but the measured value is only 1017 for the detector with a three-layer DBR. Obviously, the measured results did not achieve the quality of the theoretical values, mainly because the dielectric losses are not taken into consideration in the calculations. To analyze the effects of dielectric losses, the reflectance of the hybrid Tamm cavity is calculated with dielectric constants of the HRFZ-Si with different imaginary parts (Supplementary Note 5). The calculations show that the hybrid Tamm cavity is sensitive to the refractive index, and the resonant frequency and reflectance have a strong dependence on the permittivity of the HRFZ-Si and the metal. Moreover, the terahertz source is tuned to a resolution of 0.18 GHz in the experiment and the frequency interval used in the simulation was 0.1 GHz, which may also be why the measured $Q$ value is not as high as the calculated value.

### Tunability response of the Tamm-cavity detector

To illustrate that the resonant modes of the detectors with a hybrid Tamm cavity can be tuned by controlling the substrate thickness ($d_{substrate}$) of the detector chip, the substrate is mechanically thinned from 510 to 470 μm and then to 420 μm, and assembled with the same three Si/air layers. The measured optical voltage responsivities of these detectors are shown in Fig. 5a. As $d_{substrate}$ decreased black dashed arrow in [Fig. 5a], the resonant frequency of the detector became higher, which is consistent with our calculated results [Fig. 1i]. Within the range of measured frequencies, the resonant frequency with a substrate thickness of 510 μm corresponds to the resonant modes $N = 4$ and 5 in the substrate layer. The resonant frequency with a

substrate thickness of 470 µm corresponds to the resonant mode $N = 4$ in the substrate layer. The resonant frequency with a substrate thickness of 420 µm corresponds to the resonant modes $N = 3$ and 4. The high resonant mode in the hybrid Tamm cavity with a thick substrate overlaps with the low resonant mode with a thin substrate.

To verify the accuracy of the above design and analysis, in particular to demonstrate the tunability and overlap of the cavity modes, the measured resonant frequencies of the hybrid Tamm-cavity detector with different values of $d_{substrate}$ [red circled crosses] and the calculated resonant frequencies [extracted from Fig. 1h and shown as blue lines] are plotted in Fig. 5b. The measured and calculated values match very well. The black dashed arrow indicates the blueshifts, and the cyan region is where the cavity modes overlap. Tunable detection can be realized with this hybrid Tamm cavity just by mechanically thinning the substrate and assembling the DBR. Moreover, the signals from other bands can be filtered out by the cavity detection system. Due to the non-negligible dielectric loss and absorption in the substrate layer, the $Q$ value and bandwidth have both significantly deteriorated compared to the theoretical values [Fig. 1i]. Still, this is the pioneering report of a hybrid Tamm cavity terahertz detector, and it achieves an ultra-high resonant $Q$ value and narrow response bandwidth experimentally.

## Discussion

We demonstrated a terahertz detector integrated into a Tamm cavity. The detector chip, positioned between a multilayer Si/air DBR and an Au reflector, exhibits significantly enhanced interaction with terahertz signals within this hybrid Tamm cavity. At the resonance wavelength of the Tamm mode, the Au film and top DBR trap light effectively in the cavity, resulting in local enhancement of electric field at the detector. The detector achieves an exceptional $Q$ ($Q = 1017$) with an extraordinarily narrow bandwidth (FWHM = 469 MHz). The $Q$ of this hybrid structure surpasses that of a pure Tamm cavity and a Fabry-Perot cavity. The ability to fine-tune the frequency for a narrow bandwidth by adjusting $d_{substrate}$ makes this approach highly promising for developing terahertz spectrometers. The presented hybrid Tamm cavity, achievable through straightforward MEMS processing, stacking, and assembly without altering the device's original resonant frequency, offers a simplified design and implementation. The versatility of this hybrid Tamm-cavity terahertz detector extends to enhancing the performance of various terahertz devices, particularly in the realm of high-power sources, high-sensitivity detectors, and high-performance functional devices. Furthermore, its application holds promise for groundbreaking investigations into the strong coupling between 2D materials and terahertz waves.

## Methods
### Experimental setup and optical responsivity characterization
For the optical responsivity measurements, the detector under test was biased by a dc current (0.4 mA). The radiation was focused by two off-axis parabolic mirrors to yield the largest possible signal from the detector. For the alignment, a laser beam was used for rough adjustment, and then the detector was moved until its response voltage was a maximum. The photovoltage data were collected by a lock-in amplifier (SR830). The terahertz radiation source from 0.34 to 0.50 THz was obtained using multipliers in series (Agilent E8257D microwave source + VDI-AMC-336 + WR4.3 × 2 + WR2.2 × 2). The output power of the terahertz source was about 50 µW, which was varied with the signal frequency. It was modulated using a 4-kHz TTL signal. A thermal sensor (3A-P-THz, Ophir) was used to calibrate the optical responsivity as $R_O = V / P$, where $P$ is the total incident power and $V$ is the output voltage of the detector. To make it easier to compare and explain the responses of detectors with different cavity structures, the entire power incident on the detector was simply assumed to be effectively absorbed by the microbolometer. All measurements were performed in air at room temperature[81,82].

### Numerical simulations
TMM and electromagnetic simulation software (FDTD) are applied to calculate the reflectivity spectra associated with the profiles of the intensity enhancement of the electric field. In the simulations, the permittivity of metal Au is described using the Drude model:

$$\varepsilon(\omega) = \varepsilon_\infty + \frac{\omega_p^2}{i\omega\gamma - \omega^2}$$

where $\varepsilon_\infty = 4.8952$, $\omega_P/2\pi = 2126.4$ THz, $\gamma/2\pi = 19.6$ THz, and $n_M = \sqrt{\varepsilon(\omega)}$.

In the simulation, the refractive indices of the other materials (e.g., HRFZ-Si) were also from ref. 79.

## Data availability
The authors declare that all relevant data are available in the paper and Supplementary Information, or from the corresponding author on request.

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

## Acknowledgements

We acknowledge support from the Innovation Program for Quantum Science and Technology (No. 2021ZD0303401 to L.K.), the Fundamental Research Funds for the Central Universities, the National Natural Science Foundation of China (Grant Nos. 62271245 to X.T., 62227820 to J.C., 62271242 to X.J., 62071218 to X.J., 12033002 to L.Z., 62071214 to Q.Z., 62004093 to R.S., 62035014 to R.S., 62288101 to H.W. and 11227904 to P.W.), the National Key R&D Program of China (Grant No. 2018YFB1801504 to X.T.), the Excellent Youth Natural Science Foundation of Jiangsu Province (Grant No. BK20200060 to X.T.), the Priority Academic Program Development of Jiangsu Higher Education Institutions (PAPD), the Key Lab of Optoelectronic Devices and System with Extreme Performance and Jiangsu Key Laboratory of Advanced Techniques for Manipulating Electromagnetic Waves.

## Author contributions

X.T. and L.K. conceived the research. P.W. co-supervised the project. Y.Z., X.Y., Y.R., and X.T. performed the reflectivity and transmittivity spectra calculations. X.T., W.W., B.Y., Z.Y., C.Z., and H.S. fabricated the devices and performed the measurements. X.T. prepared the samples. Y.Z., S.Z., W.T., X.Y., and D.D. assisted in preparing the paper. X.T. wrote the paper. R.S., C.W., D.D., R.X., Q.Z., L.Z., X.J., H.W., J.C., L.K., and P.W. participated in discussions on this manuscript. Thanks to Zhanzhang Mai for his assistance in revising the manuscript. All authors discussed the results and commented on the manuscript.

## Competing interests

The authors declare no competing interests.
