## [Peer Review File · Nature Communications]

Tamm-cavity terahertz detectorREVIEWER COMMENTS

Reviewer #1 (Remarks to the Author):

Review of NCOMMS-23-22742-T

Tamm-cavity terahertz detector

Xuecou Tu^{1,2,*}, Yichen Zhang¹, Shuyu Zhou¹, Wenjing Tang¹, Xu Yan¹, Yunjie Rui¹, Wohu Wang¹, Bingnan Yan¹, Chen Zhang¹, Ziyao Ye¹, Hongkai Shi¹, Runfeng Su¹, Daxing Dong⁴, Chao Wan³, Ruiying Xu⁵, Qing-Yuan Zhao^{1,3}, La-Bao Zhang^{1,3}, Xiao-Qing Jia^{1,2}, Huabing Wang^{1,3}, Lin Kang^{1,2,*}, Jian Chen^{1,3} and Peiheng Wu^{1,2,*}

The authors study optical response of Fabry-Perot cavity and Tamm states in the THz region. The THz detector (microbolometer) embedded in to the structure which support cavity modes and optical Tamm states. Generally the idea not new, a lot of structures with very similar optical response were studied in the THz and other wavelength regions. From the manuscript is difficult to understand what kind of modes was studied for enhance the optical response: Fabry-Perot cavity, Tamm states or coupled Tamm-cavity modes. The strong light-matter interaction phenomena is used very ambiguously as well as the Tamm-cavity modes abbreviation. The authors used a lot of titles: Tamm plasmons, optical Tamm states, TP cavity, hybrid TP cavity, but the differences between them is not explained at all. I would guess in many cases it is the same modes, however, differences between TP and hybrid TP cavities remain unclear. It's not indicated, where all these modes in the spectra. In Fig.1 (h) the authors indicate the resonant modes of the cavity and Tamm cavity at certain thickness, however do not explain the splitting of cavity mode between 0.35 – 0.4 THz region, which can be important for understanding the interaction between different modes.

Also, the authors claim that it is the first time when detector is embedded in the structure and experimentally measured, however they do not explain the advantages and benefits of such solution. The applications presented in very general form without any real cases, thus difficult to judge about contribution to field of THz detection.

In spite of all noticed omissions the article can not be published.

Reviewer #2 (Remarks to the Author):

The authors present THz detectors made of Nb₅N₆ microbolometer on the substrate embedded in a hybrid Tamm cavity and demonstrate a Q value of 1017 and a bandwidth of 469 MHz for THz direct detection. These THz detectors may be of potential interest. However, I do not find the quality of the work meeting criteria for publication in Nature Communications.

First of all, the description of the cavity structure is confusing to me. The authors mention "terahertz detector integrated in a hybrid Tamm cavity". What is "hybrid" about the cavity structure presented? Is it the presence of the detector substrate? If the thickness of the substrate is a multiple of $\lambda/4n$, then it's a Tamm cavity (which is not hybrid), because the DBR+substrate is a Bragg mirror, and the cavity is therefore of zero thickness. Note that the name "Tamm" cavity refers to an analogy with Tamm electronic states, which are surface states in semiconductor structures, appearing for electronic energies in the bandgap of the material due to a periodicity break near the material surface. In addition, the authors mention: "the detector substrate is used as the defect layer of the hybrid cavity". This sentence contributes to confusing the description of the complete structure. The authors should clarify this point. In particular, they could show the electric field profile in the structure without DBR, and the reflectance of the DBR+substrate structure. Moreover, the Tamm plasmon terminology makes no sense at THz frequencies, as the plasmon resonance of gold occurs at optical wavelengths. It would be preferable to use "Tamm mode" rather than "Tamm plasmon".

The advantage of using a Tamm cavity over a THz Fabry-Perot cavity is not obvious here. In fact, the complete structure does not reduce the spatial extension of the electric field in the Y direction, and therefore does not offer the possibility of enhanced confinement. The authors should clarify the interest of their approach compared to a THz FP cavity.

Why has the optical responsivity only increased by a factor of about two in the presence of the THz cavity? The authors should comment on the weak improvement in THz detection performance. Furthermore, does the presence of the cavity improve other key parameters for photodetection ?

The discussion of the discrepancy between the experimental quality factors and theoretical predictions is not entirely convincing. Why didn't the authors indicate the quality factor extracted from the reflection spectra calculated for different imaginary parts of the refractive index of the silicon layer? These values could support the authors' claim that the discrepancy arises from dielectric losses not included in the calculation. I also recommend discussing other origins, such as inhomogeneous Tamm mode broadening due to local variations in layer thickness and DBR stack misalignment, or effects related to potential roughness at the silicon surface after etching.

I consider that the results reported in this manuscript do not properly support conclusions and claims and do not represent the type of substantial advance justifying publication in Nature Communications.

Reviewer #3 (Remarks to the Author):

Authors of the article presented the design and characterization of the novel Tamm-cavity based THz detector. The results are presented in a clear manner and are supported by theoretical and numerical analysis.

I recommend the article for the publication, however I advise the authors to address the following:

-In section 2 of the supplementary material authors discuss the condition of the Tamm-resonances, which is also given in the main-text Eq. 1. From there we can conclude that phase of the reflection from the Bragg-mirror should affect the resonance frequency (it would affect the phase and the phase then affects the frequency). However, throughout the text authors claim that frequency of the resonances is defined by the cavity and the influence of the Bragg-mirror can be omitted. Given that without the Bragg-mirror the corresponding phase is substituted by the air/cavity reflection phase - authors are advised to investigate the question and clarify why the phase of Bragg-mirror reflection does not have a strong effect on the resonance frequencies.

-While comparing the results of numerical modeling with the experiment authors claim that absence of responsivity increase between structures with single and triple Si/air layers on top can be attributed to the light absorption. It however somewhat contradicts the findings in section 4 of the supplementary material where (see Figure S9) the reflection deep becomes stronger for higher absorption (higher imaginary part of the refractive index) for some frequencies (see around 0.4 THz) and weaker for others (see around 0.47 THz). I recommend authors to do calculation of the field at the detector position for varying absorption (imaginary part of the refractive index) to give a fuller picture.

Reviewer # 1:

The authors study optical response of Fabry-Perot cavity and Tamm states in the THz region. The THz detector (microbolemeter) embedded in to the structure which support cavity modes and optical Tamm states. Generally the idea not new, a lot of structures with very similar optical response were studied in the THz and other wavelenght regions.

Response: We greatly thank the reviewer for carefully reviewing our manuscript. Fortunately, the reviewer provided us with many helpful suggestions to improve the manuscript. We have a high respect for the reviewer. As pointed out by the reviewer, enhanced structures based on a Tamm cavity with a built-in distributed Bragg reflector (DBR) are commonly used in photodetectors and lasers. Tamm cavities have had an important role in enhancing the interaction between material and light to realize high Q and tunable devices. These excellent properties are necessary for terahertz-band devices. However, the functional components integrated with the DBR in the terahertz spectral range have rarely been reported in the literature. The main difficulty is that the smallest planar features are of the order of $\lambda/4n_r \approx 10 \mu\text{m}$, where n_r is the refractive index of the dielectric. Terahertz wavelengths are in the range 10 to 1000 μm , so depositing thin films of an optical dielectric, which is commonly used for optical devices, cannot be used to construct microcavities, which is necessary for the DBR structures used in terahertz devices. Note that the features of terahertz microcavities are of the order of the thickness of the substrates of the terahertz devices, so researchers have also begun to use the substrates as FP cavities when constructing electromagnetic confinement devices. Obviously, the performance of these devices can be further improved if the Tamm cavity is used in optical wavebands. An optical DBR cavity composed of multiple layers of Si and air was fabricated by an ingenious and complex process. It has a very high refractive index contrast and very high Q value. Obviously, the difficulties in depositing dielectric thin films and lateral etching are hard to address at the micro-scale in the terahertz band. Simon Messelot et al. reported the Tamm activity in the terahertz band experimentally by alternately stacking thin silicon layers and air (*ACS Photonics* 7(10), 2906-2914, 2020). **The integration of Tamm cavities and functional devices needs to consider compatibility of chip preparation process, and such integrated devices have hardly been reported so far. Here, we are the first to report a Tamm cavity integrated terahertz detector.** We demonstrated experimentally that the Hybrid Tamm-cavity structure coupled detector has high Q and very narrowband optical responsivity in the terahertz band. It provides a general operating platform for other devices that need enhanced interactions between matter and a terahertz wave. In particular, it can be used to study the electronics and optoelectronics of 2D materials or to fabricate terahertz lasers, terahertz detectors, and other high-performance functional

devices.

From the manuscript is difficult to understand what kind of modes was studied for enhance the optical response: Fabry-Perot cavity, Tamm states or coupled Tamm-cavity modes. The strong light-matter interaction phenomena is used very ambiguously as well as the Tamm-cavity modes abbreviation. The authors used a lot of titles: Tamm plasmons, optical Tamm states, TP cavity, hybrid TP cavity, but the differences between them is not explained at all. I would guess in many cases it is the same modes, however, differences between TP and hybrid TP cavities remain unclear. It's not indicated, where all these modes in the spectra.

Response: Thank you for highlighting the title confusion. We have opted to use "hybrid Tamm-cavity" to encompass the entire configuration, stimulating coupled Tamm-cavity modes or hybrid modes. The rationale behind this decision is outlined below. A pure Tamm structure comprises a DBR and a metal reflector, exciting a Tamm state within the DBR stopband, as depicted in Fig. R1(a). Hybrid Tamm structure, which is formed by sandwiching a dielectric layer between DBR and metal layer, excites a much higher Q than a typical Tamm structure by coupling the Tamm mode and cavity mode [1]. On top of that, we ingeniously use the substrate of the THz detector as cavity to construct a hybrid Tamm cavity (HTC) as shown in Fig. R1(b). The HTC has not only all the advantages of the hybrid Tamm structure such as high Q and frequency tunability, but also a simplified fabrication process. As illustrated in Fig. R1(b), the cavity is sufficiently thick to excite FP cavity modes, which couple with the Tamm mode, resulting in multiple hybrid modes. This is evidenced by the substantial overlap between the calculation of cavity modes and the reflection spectrum of the hybrid structure with varying substrate thickness of the detector. These multiple modes exhibit a much higher Q (2907 at 0.65 THz) than that of the pure Tamm mode (264 at 0.65 THz). Fig. R1(c) demonstrates the simulated electrical intensity of FP cavity, pure Tamm cavity (point A in Fig. R1(a)), hybrid Tamm cavity (point B in Fig. R1(b)), hybrid Tamm cavity (point C in Fig. R1(b)), which illustrates that the hybrid Tamm cavity effectively confines the electrical field within the cavity, showcasing maximum electrical intensity at the detector surface.

As pointed out by the reviewer, we have decided to standardized our usage to refer to our structure as the 'hybrid Tamm-cavity'. Concurrently, we use 'hybrid Tamm cavity mode' to denote the modes excited within our structure. Additionally, the pertinent information regarding this nomenclature has been incorporated into the third paragraph of the revised manuscript's introduction. It reads:

“Hybrid Tamm cavity (HTC), which is formed by sandwiching a dielectric layer between DBR and metal layer, excites a much higher Q than a typical Tamm cavity by

coupling the Tamm mode and FP mode [56]. On top of that, we ingeniously use the substrate of the THz detector as the FP cavity to construct the HTC as shown in Fig. 1(a). The HTC has not only all the advantages of the HTC such as high Q and frequency tunability, but also a simplified fabrication process.”

[1] Zhiyu Wang, J. Kenji Clark, Ya-Lun Ho, Bertrand Vilquin, Hirofumi Daiguji, and Jean-Jacques Delaunay, Narrowband Thermal Emission Realized through the Coupling of Cavity and Tamm Plasmon Resonances, *ACS Photonics*, 5 (6), 2446-2452 (2018).

[2] Lydie Ferrier, Hai Son Nguyen, Cécile Jamois, Lotfi Berguiga, Clémentine Symonds, Joël Bellessa, and Taha Benyattou, Tamm plasmon photonic crystals: From bandgap engineering to defect cavity, *APL Photonics* 4, 106101 (2019).

[3] Simon Messelot, Clémentine Symonds, Joël Bellessa, Jérôme Tignon, Sukhdeep Dhillon, Jean-Blaise Brubach, Pascale Roy, and Juliette Mangeney, Tamm Cavity in the THz Spectral Range, *ACS Photonics* 7 (10), 2906-2914 (2020).

Fig. R1 (a) Simulated reflection spectrum of a pure Tamm structure consists of 3.5 pairs Si/air DBR covered with Au layer. (b) Simulation reflection spectrum of a designed Hybrid Tamm-cavity which consists of a 510 μm Si substrate layer inserted between 3 pairs Si/air DBR and an Au layer. (c) From left to right are the simulated electrical intensity distribution of FP cavity, pure Tamm cavity (point A in Fig. R1(a)), hybrid Tamm cavity (point B in Fig. R1(b)), hybrid Tamm cavity (point C in Fig. R1(b)).

In Fig.1 (h) the authors indicate the resonant modes of the cavity and Tamm cavity at certain thickness, however do not explain the splinting of cavity mode between 0.35 – 0.4 THz region, which can be important for understanding the interaction between different modes.

Response: In Fig. R2(a), the frequency range of 0.35 THz to 0.4 THz lies on the edge of the DBR stopband, showcasing splintering across all modes within this range. This suggests that the root cause may stem from the characteristics of the DBR. The phase representation of the DBR is illustrated by the yellow line in Fig. R2(a). Notably, when the frequency falls below 0.4 THz (at point C) and surpasses 0.37 THz, the DBR phase exhibits a deviation from 0, exerting a negligible impact on the resonant frequency. Within the interval of 0.37 THz (point B) to 0.35 THz (point A), the DBR phase progressively approaches 0. Consequently, the resonant frequency diverges from the cavity mode frequency between points C and B, then reconverges between points B and A. The observed reduction in structural absorption can be attributed to the DBR's reflective properties. Beyond the DBR stopband, reflections become imperfect, leading to the leakiness of the hybrid Tamm-cavity modes [1]. As depicted in Fig. R2(a), the reflection experiences a decline from point C to point B, resulting in a subdued line in Fig. R2(b). Subsequently, from point B to point A, there is a resurgence in DBR reflection, hence a faint but discernible line spanning from 0.26 THz to 0.35 THz.

We have included an explanation about the splinting in the seventh paragraph in the section of Device design, and it reads:

“The splintering observed within the 0.35 THz - 0.4 THz region is due to the deviation of the DBR phase from 0 between 0.37 THz and 0.4 THz. This deviation leads to a shift in the resonant frequency away from the cavity mode frequency. Moreover, the reflection of the DBR diminishes within this range as it approaches the edge of the stopband, resulting in imperfections. Consequently, the hybrid mode exhibits leakiness, leading to lower electrical intensity.”

[1] Zhiyu Wang, J. Kenji Clark, Ya-Lun Ho, Bertrand Vilquin, Hirofumi Daiguji, and Jean-Jacques Delaunay, Narrowband Thermal Emission Realized through the Coupling of Cavity and Tamm Plasmon Resonances, *ACS Photonics*, 5 (6), 2446-2452 (2018)

[2] Lydie Ferrier, Hai Son Nguyen, Cécile Jamois, Lotfi Berguiga, Clémentine Symonds, Joël Bellessa, and Taha Benyattou, Tamm plasmon photonic crystals: From bandgap engineering to defect cavity, *APL Photonics* 4, 106101 (2019).

[3] Simon Messelot, Clémentine Symonds, Joël Bellessa, Jérôme Tignon, Sukhdeep Dhillon, Jean-Blaise Brubach, Pascale Roy, and Juliette Mangeney, Tamm Cavity in the THz Spectral Range, *ACS Photonics* 7 (10), 2906-2914 (2020).

Fig. R2 (a) Calculated reflection spectrum and the phase of the designed 3.5 pairs Si/air DBR. (b) Relation between the electric field (E_d) and substrate thickness of the detector chip ($d_{\text{substrate}}$) for a hybrid Tamm cavity in the range 0.25–0.6 THz.

Also, the authors claim that it is the first time when detector is embedded in the structure and experimentally measured, however they do not explain the advantages and benefits of such solution.

Response: The entire cavity, devoid of the detector, exclusively utilizes Silicon, allowing fabrication through well-established MEMS processes. Additionally, our innovative approach integrates the detector's substrate to the hybrid Tamm cavity, streamlining fabrication steps. Leveraging this design, the pre-existing detector benefits from this hybridization, necessitating only a metal layer deposition on the backside of the substrate, as both the DBR and detector chip are independently fabricated. At last, the DBR block was bonded to the detector chip to form the hybrid Tamm coupled detector (**FIG. S7, See Supplementary Note S3**). Moreover, fine-tuning the resonant frequency within the DBR stopband merely requires adjusting the cavity's thickness.

FIG. S7. Fabricating the hybrid Tamm-cavity detector.

Compared to a pure Tamm structure, our hybrid Tamm cavity design requires fewer fabrication steps while achieving significantly higher Q . In contrast, obtaining a Fabry-Perot (F-P) cavity with a substantial absorptance peak and a narrow bandwidth proves challenging [1]. Additionally, our hybrid Tamm cavity surpasses reported Q of THz F-P cavities [2-3]. With advancements in MEMS technology and low-loss Silicon, the Q of our hybrid Tamm cavity could potentially reach ~ 2000 in the future.

This simplified approach, utilizing MEMS processing, stacking, and assembly without altering the device's original resonant frequency, streamlines design and implementation. The versatility of this hybrid Tamm-cavity terahertz detector extends to enhancing the performance of various terahertz devices, particularly in high-power sources, high-sensitivity detectors, and high-performance functional devices. Furthermore, its application holds promise for breakthroughs in exploring the strong coupling between 2D materials and terahertz waves.

[1] Zhiyu Wang, J. Kenji Clark, Ya-Lun Ho, Bertrand Vilquin, Hirofumi Daiguji, and Jean-Jacques Delaunay, Narrowband Thermal Emission Realized through the Coupling of Cavity and Tamm Plasmon Resonances, *ACS Photonics*, 5 (6), 2446-2452 (2018)

[2] Lydie Ferrier, Hai Son Nguyen, Cécile Jamois, Lotfi Berguiga, Clémentine Symonds, Joël Bellessa, and Taha Benyattou, Tamm plasmon photonic crystals: From bandgap engineering to defect cavity, *APL Photonics* 4, 106101 (2019).

[3] Simon Messelot, Clémentine Symonds, Joël Bellessa, Jérôme Tignon, Sukhdeep Dhillon, Jean-Blaise Brubach, Pascale Roy, and Juliette Mangeney, Tamm Cavity in the THz Spectral Range, *ACS Photonics* 7 (10), 2906-2914 (2020).

[4] Zhang, Q.; Lou, M.; Li, X.; Reno, J. L.; Pan, W.; Watson, J. D.; Manfra, M. J.; Kono, J. Collective non-perturbative coupling of 2D electrons with high-quality-factor terahertz cavity photons. *Nat. Phys*, 12, 1005–1011 (2016)

[5] Meng, F.; Thomson, M. D.; Klug, B.; Čibiraitė, D.; Ul-Islam, Q.; Roskos, H. G. Nonlocal collective ultrastrong interaction of plasmonic metamaterials and photons in a terahertz photonic crystal cavity, *Opt. Express*, 27, 24455 (2019)

We have concluded the advantage in the conclusion and discussion section in the revised manuscript. It reads:

“We present a pioneering demonstration of a terahertz detector integrated into a hybrid Tamm cavity. The detector chip, positioned between a multilayer Si/air DBR and an Au reflector, exhibits significantly enhanced interaction with terahertz signals within this hybrid Tamm cavity. As a result, the detector achieves an exceptional Q ($Q = 1017$) with an extraordinarily narrow bandwidth (FWHM = 469 MHz). Notably, the Q of this hybrid Tamm cavity surpasses that of a typical THz Tamm structure (over 200) [33] and

a Fabry-Perot cavity (over 200) [83, 84]. The ability to fine-tune the frequency for a narrow bandwidth by adjusting $d_{\text{substrate}}$ makes this approach highly promising for developing terahertz spectrometers. Our innovative hybrid Tamm cavity, achievable through straightforward MEMS processing, stacking, and assembly without altering the device's original resonant frequency, offers a simplified design and implementation. The versatility of this hybrid Tamm cavity terahertz detector extends to enhancing the performance of various terahertz devices, particularly in the realm of high-power sources, high-sensitivity detectors, and high-performance functional devices. Furthermore, its application holds promise for groundbreaking investigations into the strong coupling between 2D materials and terahertz waves.”

Finally, we thank the reviewer again for the time spent on our manuscript and providing us with many useful suggestions. We hope that in the revised manuscript, we have addressed the reviewer’s concern in a manner that is somewhat satisfactory.

Reviewer # 2:

The authors present THz detectors made of Nb5N6 microbolometer on the substrate embedded in a hybrid Tamm cavity and demonstrate a Q value of 1017 and a bandwidth of 469 MHz for THz direct detection. These THz detectors may be of potential interest. However, I do not find the quality of the work meeting criteria for publication in Nature Communications.

Response: We greatly thank the reviewer for carefully reviewing our manuscript. Fortunately, the reviewer provided us with many helpful suggestions to improve the manuscript. Following the reviewer’s suggestion, in the revised manuscript, the related content had been redrafted. We have tried our best effort to improve the readability. We hope that the revised manuscript, although still being far away from perfection, will not distract the readers much from obtaining an acceptable understanding.

First of all, the description of the cavity structure is confusing to me. The authors mention "terahertz detector integrated in a hybrid Tamm cavity". What is "hybrid" about the cavity structure presented? Is it the presence of the detector substrate? If the thickness of the substrate is a multiple of $\lambda/4n$, then it's a Tamm cavity (which is not hybrid), because the DBR + substrate is a Bragg mirror, and the cavity is therefore of zero thickness. Note that the name "Tamm" cavity refers to an analogy with Tamm electronic states, which are surface states in semiconductor structures, appearing for electronic energies in the bandgap of the material due to a periodicity break near the material surface. In addition, the authors mention: "the detector substrate is used as

the defect layer of the hybrid cavity". This sentence contributes to confusing the description of the complete structure. The authors should clarify this point. In particular, they could show the electric field profile in the structure without DBR, and the reflectance of the DBR + substrate structure.

Response: Thank you for highlighting the title confusion. We have opted to use "hybrid Tamm-cavity" to encompass the entire configuration, stimulating coupled Tamm-cavity modes or hybrid modes. The rationale behind this decision is outlined below. A pure Tamm structure comprises a DBR and a metal reflector, exciting a Tamm state within the DBR stopband, as depicted in Fig. R1(a). Hybrid Tamm structure, which is formed by sandwiching a dielectric layer between DBR and metal layer, excites a much higher Q than a typical Tamm structure by coupling the Tamm mode and cavity mode [1]. On top of that, we ingeniously use the substrate of the THz detector as cavity to construct a Hybrid Tamm Cavity (HTC) as shown in Fig. R1(b). The HTC has not only all the advantages of the hybrid Tamm structure such as high Q and frequency tunability, but also a simplified fabrication process. As illustrated in Fig. R1(b), the cavity is sufficiently thick to excite cavity modes, which couple with the Tamm mode, resulting in multiple hybrid modes. This is evidenced by the substantial overlap between the calculation of cavity modes and the reflection spectrum of the hybrid Tamm cavity with varying cavity thickness. These multiple modes exhibit a much higher Q (2907 at 0.65 THz) than that of the pure Tamm mode (264 at 0.65 THz). Fig. R2(c) demonstrates the simulated electrical intensity of FP cavity, pure Tamm cavity (point A in Fig. R2(a)), hybrid Tamm cavity (point B in Fig. R2(b)), hybrid Tamm cavity (point C in Fig. R2(b)), which illustrates that the hybrid Tamm cavity effectively confines the electrical field within the substrate, showcasing maximum electrical intensity at the detector surface.

We have decided to standardized our usage to refer to our structure as the 'hybrid Tamm cavity '. Concurrently, we use 'hybrid Tamm-cavity mode' to denote the modes excited within our structure. Additionally, the pertinent information regarding this nomenclature has been incorporated into the third paragraph of the revised manuscript's introduction. It reads:

“Hybrid Tamm cavity (HTC), which is formed by sandwiching a dielectric layer between DBR and metal layer, excites a much higher Q than a typical Tamm cavity by coupling the Tamm mode and FP mode [56]. On top of that, we ingeniously use the substrate of the THz detector as the FP cavity to construct the HTC as shown in Fig. 1(a). The HTC has not only all the advantages of the HTC such as high Q and frequency tunability, but also a simplified fabrication process.”

[1] Zhiyu Wang, J. Kenji Clark, Ya-Lun Ho, Bertrand Vilquin, Hirofumi Daiguji, and

Jean-Jacques Delaunay, Narrowband Thermal Emission Realized through the Coupling of Cavity and Tamm Plasmon Resonances, ACS Photonics, 5 (6), 2446-2452 (2018).

[2] Lydie Ferrier, Hai Son Nguyen, Cécile Jamois, Lotfi Berguiga, Clémentine Symonds, Joël Bellessa, and Taha Benyattou, Tamm plasmon photonic crystals: From bandgap engineering to defect cavity, APL Photonics 4, 106101 (2019).

[3] Simon Messelot, Clémentine Symonds, Joël Bellessa, Jérôme Tignon, Sukhdeep Dhillon, Jean-Blaise Brubach, Pascale Roy, and Juliette Mangeney, Tamm Cavity in the THz Spectral Range, ACS Photonics 7 (10), 2906-2914 (2020).

Fig. R3 (a) Simulated reflection spectrum of a pure Tamm structure consists of 3.5 pairs Si/air DBR covered with Au layer. (b) Simulation reflection spectrum of a designed hybrid Tamm cavity which consists of a 510 μm Si layer inserted between 3 pairs Si/air DBR and an Au layer. (c) From left to right are the simulated electrical intensity distribution of FP cavity, pure Tamm cavity (point A in Fig. R2(a)), hybrid Tamm cavity (point B in Fig. R2(b)), hybrid Tamm cavity (point C in Fig. R2(b)). (d) Simulation reflection spectrum of a designed hybrid Tamm cavity without the Au reflector.

Moreover, the Tamm plasmon terminology makes no sense at THz frequencies, as the plasmon resonance of gold occurs at optical wavelengths. It would be preferable to use "Tamm mode" rather than "Tamm plasmon".

Response: Thank you for your invaluable advice. As per your suggestion, we've removed the term "Tamm plasmon". Instead, we've adopted the term "hybrid Tamm-cavity mode" to represent the enhancement mode within our hybrid Tamm cavity, as

discussed previously.

The advantage of using a Tamm cavity over a THz Fabry-Perot cavity is not obvious here. In fact, the complete structure does not reduce the spatial extension of the electric field in the Y direction, and therefore does not offer the possibility of enhanced confinement. The authors should clarify the interest of their approach compared to a THz FP cavity.

Response: The entire hybrid Tamm cavity, devoid of the detector, exclusively utilizes Silicon, allowing fabrication through well-established MEMS processes. Additionally, our innovative approach integrates the detector's substrate into the hybrid Tamm cavity, streamlining fabrication steps. Leveraging this design, the pre-existing detector benefits from this hybridization, necessitating only a metal layer deposition on the substrate's back, as both the DBR and detector chip are independently fabricated. Moreover, fine-tuning the resonant frequency within the DBR stopband merely requires adjusting the cavity's thickness.

Compared to a pure Tamm structure, our hybrid design requires fewer fabrication steps while achieving significantly higher Q . In contrast, obtaining a Fabry-Perot (F-P) cavity with a substantial absorptance peak and a narrow bandwidth proves challenging [1]. Additionally, our hybrid Tamm cavity surpasses reported Q of THz F-P cavities [2-3]. With advancements in MEMS technology and low-loss Silicon, the Q of our hybrid Tamm cavity could potentially reach ~ 2000 in the future.

This simplified approach, utilizing MEMS processing, stacking, and assembly without altering the device's original resonant frequency, streamlines design and implementation. The versatility of this hybrid Tamm-cavity terahertz detector extends to enhancing the performance of various terahertz devices, particularly in high-power sources, high-sensitivity detectors, and high-performance functional devices. Furthermore, its application holds promise for breakthroughs in exploring the strong coupling between 2D materials and terahertz waves.

[1] Zhiyu Wang, J. Kenji Clark, Ya-Lun Ho, Bertrand Vilquin, Hirofumi Daiguji, and Jean-Jacques Delaunay, Narrowband Thermal Emission Realized through the Coupling of Cavity and Tamm Plasmon Resonances, ACS Photonics, 5 (6), 2446-2452 (2018)

[2] Lydie Ferrier, Hai Son Nguyen, Cécile Jamois, Lotfi Berguiga, Clémentine Symonds, Joël Bellessa, and Taha Benyattou, Tamm plasmon photonic crystals: From bandgap engineering to defect cavity, APL Photonics 4, 106101 (2019).

[3] Simon Messelot, Clémentine Symonds, Joël Bellessa, Jérôme Tignon, Sukhdeep Dhillon, Jean-Blaise Brubach, Pascale Roy, and Juliette Mangeney, Tamm Cavity in the

THz Spectral Range, ACS Photonics 7 (10), 2906-2914 (2020).

[4] Zhang, Q.; Lou, M.; Li, X.; Reno, J. L.; Pan, W.; Watson, J. D.; Manfra, M. J.; Kono, J. Collective non-perturbative coupling of 2D electrons with high-quality-factor terahertz cavity photons. Nat. Phys, 12, 1005–1011 (2016)

[5] Meng, F.; Thomson, M. D.; Klug, B.; Čibiraitė, D.; Ul-Islam, Q.; Roskos, H. G. Nonlocal collective ultrastrong interaction of plasmonic metamaterials and photons in a terahertz photonic crystal cavity, Opt. Express, 27, 24455 (2019)

We have concluded the advantage in the conclusion and discussion section in the revised manuscript. It reads:

“We present a pioneering demonstration of a terahertz detector integrated into a hybrid Tamm cavity. The detector chip, positioned between a multilayer Si/air DBR and an Au reflector, exhibits significantly enhanced interaction with terahertz signals within this hybrid Tamm cavity. As a result, the detector achieves an exceptional Q ($Q = 1017$) with an extraordinarily narrow bandwidth (FWHM = 469 MHz). Notably, the Q of this hybrid Tamm cavity surpasses that of a typical THz Tamm cavity (over 200) [33] and a Fabry-Perot cavity (over 200) [83, 84]. The ability to fine-tune the frequency for a narrow bandwidth by adjusting d_{cavity} makes this approach highly promising for developing terahertz spectrometers. Our innovative hybrid Tamm cavity, achievable through straightforward MEMS processing, stacking, and assembly without altering the device's original resonant frequency, offers a simplified design and implementation. The versatility of this hybrid Tamm-cavity terahertz detector extends to enhancing the performance of various terahertz devices, particularly in the realm of high-power sources, high-sensitivity detectors, and high-performance functional devices. Furthermore, its application holds promise for groundbreaking investigations into the strong coupling between 2D materials and terahertz waves.”

Why has the optical responsivity only increased by a factor of about two in the presence of the THz cavity? The authors should comment on the weak improvement in THz detection performance. Furthermore, does the presence of the cavity improve other key parameters for photodetection? The discussion of the discrepancy between the experimental quality factors and theoretical predictions is not entirely convincing. Why didn't the authors indicate the quality factor extracted from the reflection spectra calculated for different imaginary parts of the refractive index of the silicon layer? These values could support the authors' claim that the discrepancy arises from dielectric losses not included in the calculation. I also recommend discussing other origins, such as inhomogeneous Tamm mode broadening due to local variations in layer thickness and DBR stack misalignment, or effects related to potential roughness at the

silicon surface after etching.

Response: We extracted the quality factor (Q) from the reflection spectrum of the hybrid Tamm cavity, employing different Si/air DBR pairs at 0.48 THz, depicted by the purple line in Fig. R4. When compared to the Q extracted from electrical intensity, the Q derived from the reflection spectrum aligns more closely with the experimental results. However, given that this is the inaugural integration of a 2D THz detector into such a structure, the reasons for its performance falling short of expectations may be multifaceted. Potential factors include the non-zero imaginary part of silicon, deviations in thickness due to MEMS, likely surface roughness post-etching on the silicon, and misalignment within the DBR stack.

To delve into this, we conducted calculations on the reflection spectra using Transfer Matrix Method (TMM), varying the imaginary part of silicon (k_{Si}), and extracted the corresponding Q values. As illustrated in Fig. R4(a), even a slight fluctuation in k_{Si} can precipitate a notable decline in the quality factor. Remarkably, our calculated Q closely matches the experimental data when the imaginary part of Si is set at 0.0008. Additionally, we deliberately set the absorption coefficients (α) of both the DBR and the substrate layer as zero. The influence of α on Q primarily arises from the substrate layer as the electric field remains confined within it. Notably, despite achieving only half of the theoretically anticipated Q in the experiment, it still surpasses that of a typical Tamm structure and a Fabry-Perot cavity. Furthermore, deviations in thicknesses can occur during the fabrication process. To gauge this impact, we run simulations with variations in layer thicknesses within $\pm 3 \mu\text{m}$ using TMM to compute Q . This extensive simulation conducted a million times on each DBR pair ($k_{\text{Si}} = 0$ and $k_{\text{Si}} = 0.0008$) is illustrated in Fig. R4(a), revealing that $\pm 3 \mu\text{m}$ thickness deviations can result in up to a 7% fluctuation in Q . Interestingly, with more DBR pairs or higher k_{Si} , the impact of thickness deviation diminishes, as the incident light gets absorbed (higher k_{Si}) or blocked (more DBR pairs), thereby weakening the coupling strength. Moreover, the MEMS process might introduce surface roughness at each interface, potentially leading to diffuse reflections and broadening of the resonant peak. Additionally, DBR stack misalignment, being assembled manually, can influence Q . Nonetheless, considering the pixel size set at $5 \text{ mm} \times 5 \text{ mm}$, significantly larger than the incident light's wavelength, we assume any probable effects to be limited. In conclusion, enhancing the performance of the hybrid stack in the future can be achieved through the utilization of higher-quality, low-loss silicon, and improved MEMS techniques.

Fig. R4 (a) Comparison of measured and calculated quality factor at 0.48 THz of the hybrid Tamm cavity with different pair number of DBR and imaginary part of silicon index. Error includes the thicknesses' variation of each layer is only calculated with $n = 3.4147$ and $n = 3.4147 - 0.0008i$. (b) Comparison of measured and calculated quality factor at 0.48 THz of the hybrid Tamm cavity with different pair number of DBR and $k_{Si} = 0.0008$ in cavity and DBR, respectively.

We have included more discussion in the revised manuscript in the second paragraph in Experimental result and discussion and the last paragraph in supplementary material. They read:

Second paragraph in Experimental result:

“The escalation of dielectric loss is primarily attributed to the increasing number of DBR layers. Furthermore, deviations in the thickness of each layer and surface roughness stemming from the MEMS process notably hinder the hybrid Tamm cavity's performance. In our simulations, a mere $\pm 3 \mu\text{m}$ fluctuation in layer thicknesses resulted in up to a 7% deviation in Q . Potential roughness on the silicon surface induces diffuse reflections, limiting the amount of light that can be coupled to Au, ultimately leading to a diminished Q and electrical intensity.”

Last paragraph in supplementary material:

“The Q of the hybrid Tamm cavity is adversely affected by an increased imaginary part of silicon, particularly within the substrater layer. Notably, when employing an imaginary part of silicon (k_{Si}) value of 0.0008, the measured Q closely aligns with the theoretical Q calculated using TMM.”

To better demonstrate the influence of the silicon refractive index, we have updated the figures. Fig.4, Fig. S8, and Fig. S9 have been replaced with Fig. R5, Fig. R6, and Fig. R7, as outlined below for clearer illustration.

FIG. R5 Comparison of measured and calculated Q values at a resonant mode (0.48 THz) of the detector with different numbers of layers. The calculated Q values are extracted from reflection spectrum with $n_{Si} = 3.4147$ and $n_{Si} = 3.4147 - 0.0008i$, respectively.

FIG. R6 Calculated reflection spectra for different real parts of the refractive index of HRFZ-Si layer at $d_{substrate} = 510 \mu m$: (a) $n_H = 3.4147$, (b) $n_H = 3.42$, and (c) $n_H = 3.45$. (d) Calculated Q values at a resonant mode (around 0.48 THz) of the detector with different numbers of layers with $n_{Si} = 3.4147$, $n_{Si} = 3.42$ and $n_{Si} = 3.45$

Fig. R7 Calculated reflection spectra for different imaginary parts of the refractive index of the silicon layer at $d_{substrate} = 510 \mu\text{m}$: (a) $n_{Si} = 3.4147 - 0.001i$, (b) $n_{Si} = 3.4147 - 0.0008i$, (c) $n_{Si} = 3.4147 - 0.00008i$. (d) Calculated Q values at a resonant mode (0.48 THz) of the detector with different numbers of layers with $k_{Si} = 0$, $k_{Si} = 0.0008$, k_{Si} of the substrate = 0.0008 and k_{Si} of the DBR = 0.0008.

Finally, we thank the reviewer again for spending his/her effort reviewing our manuscript. In particular, we greatly thank the reviewer for providing us with many helpful suggestions. We hope that in the revised manuscript, we have addressed the reviewer's concern in a manner that is somewhat satisfactory.

Reviewer # 3:

Authors of the article presented the design and characterization of the novel Tamm-cavity based THz detector. The results are presented in a clear manner and are supported by theoretical and numerical analysis. I recommend the article for the publication.

Response: We greatly thank the reviewer for his/her appreciation of the novelty and impact of our manuscript. In particular, we greatly thank the reviewer for providing us with many helpful suggestions.

In section 2 of the supplementary material authors discuss the condition of the Tamm-

resonances, which is also given in the main-text Eq. 1. From there we can conclude that phase of the reflection from the Bragg-mirror should affect the resonance frequency (it would affect the phase and the phase then affects the frequency). However, throughout the text authors claim that frequency of the resonances is defined by the cavity and the influence of the Bragg-mirror can be omitted. Given that without the Bragg-mirror the corresponding phase is substituted by the air/cavity reflection phase - authors are advised to investigate the question and clarify why the phase of Bragg-mirror reflection does not have a strong effect on the resonance frequencies.

Response: The phase of the reflection from the distributed Bragg reflector (DBR) indeed plays a role in affecting the resonance frequency. Illustrated in Fig. R8(a), the phase of the DBR remains close to zero, indicating a minimal impact on the resonant frequency of the hybrid Tamm cavity. However, when the frequency deviates from the center of the DBR stopband, the phase significantly shifts from zero, causing a discrepancy between the resonant frequency and the frequency of cavity modes. For instance, at frequencies such as 0.4 THz and 0.85 THz, positioned at the edge of the DBR stopband, this deviation becomes apparent. Fig. R8(b) highlights a significant overlap between the cavity modes and the hybrid modes, emphasizing that the resonant frequencies of hybrid modes are predominantly influenced by the cavity modes, specifically the thickness of the cavity.

Fig. R8 (a) Calculated reflection spectrum and the corresponding phase of DBR using TMM. (b) Simulated reflection spectrum versus the thickness of substrate layer and frequency. The white lines are different cavity modes calculated with $d_{\text{substrate}} = \frac{(2N+1)\lambda}{4n}$.

While comparing the results of numerical modeling with the experiment authors claim that absence of responsivity increase between structures with single and triple Si/air layers on top can be attributed to the light absorption. It however somewhat contradicts the findings in section 4 of the supplementary material where (see Figure S9) the reflection deep becomes stronger for higher absorption (higher imaginary part of the

refractive index) for some frequencies (see around 0.4 THz) and weaker for others (see around 0.47 THz). I recommend authors to do calculation of the field at the detector position for varying absorption (imaginary part of the refractive index) to give a fuller picture.

Response: Fig. R9 (a)(b) illustrate that as the imaginary part of silicon increases, absorption within the silicon layers intensifies, while the absorption within the Au layer diminishes at 0.4 THz and 0.48 THz. This scenario indicates reduced light coupling to the Au layer to excite hybrid modes, thereby indicating a decrease in electrical intensity at both frequencies. At 0.4 THz, positioned at the DBR stopband's edge, where the DBR reflectance is imperfect, a leaky hybrid Tamm-cavity mode emerges, resulting in lower electrical intensity and a lower Q . This leakiness leads to a slower decrease in absorption within the Au layer compared to the increasing absorption in Si, thereby causing a more pronounced reflection dip at 0.4 THz. Conversely, at 0.48 THz, the DBR exhibits perfect reflection, and the coupling between incident light and metal is robust. However, as the incident light gets absorbed in the Si layers with a higher imaginary part of silicon, the coupling strength diminishes rapidly, leading to a shallower reflection dip. Additionally, we conduct calculations on the reflection spectrum of the hybrid Tamm cavity utilizing a single pair Si/air DBR, which displays an imperfect reflectance spectrum even within the stopband. Consequently, all modes become leaky, resulting in deeper resonant dips with an increased imaginary part of silicon, as depicted in Fig. R6(e). Notably, in Fig. R6(d) and (e), the reflection spectra with $n = 3.4147 - 0.001i$ have been manually shifted by 0.1 THz for better comparison.

Fig. R9 (a) Simulated absorption of Silicon and Au layer in the designed hybrid Tamm cavity at 0.4 THz. (b) Simulated absorption of Silicon and Au layer in the designed hybrid Tamm cavity at 0.48 THz. (d) Simulated electrical intensity at device interface versus imaginary part of Silicon refractive

index at 0.4 THz and 0.48 THz respectively. (d) Simulated reflection of the hybrid Tamm cavity (3 pairs Si/air DBR) with different refractive index of silicon (blue: $n=3.4147$, orange: $n=3.4147-0.001i$). (e) Simulated reflection of hybrid Tamm cavity (1 pair Si/air DBR) with different refractive index of silicon (blue: $n= 3.4147$, orange: $n= 3.4147-0.001i$).

Finally, we thank the reviewer again for spending his/her effort reviewing our manuscript. In particular, we greatly thank the reviewer for providing us with many helpful suggestions. We hope that in the revised manuscript, we have addressed the reviewer's concern in a manner that is somewhat satisfactory.

REVIEWER COMMENTS

Reviewer #1 (Remarks to the Author):

I want to thank authors for the answers. The authors clarify that Fabry-Perot mode and Tamm optical state should form new hybrid mode what they called hybrid Tamm cavity. In the Fig.1 (b) and (h) showed resonant modes which according to authors corresponds for the hybrid Tamm-Cavity mode, however, at the substrate thickness $d=510$ um in both pictures (b) and (h) from the optical response I do not see any evidence of hybridization. Meanwhile, for about 440 um thickness (0.35-0.4 THz range) a hybridization can be clearly seen and anti-crossing effect of some cavity mode (as I guess) with photonic band edge of the distributed Bragg reflector. If authors claim that this is hybrid mode, then in what kind of coupling regime this mode is? For me the evidence of hybridization of Tamm state and cavity mode is still unclear.

Reviewer #3 (Remarks to the Author):

Authors carefully addressed the points raised in the previous review iteration. I therefore thank the authors and recommend the article for publication.

Dear Editor and Reviewers:

We would like to thank you for your efforts on our manuscripts. We greatly appreciate the constructive and insightful comments from the Editor and Reviewers. In this letter, we have carefully addressed the comments and queries raised by Reviewer 1 (comments from the referees are in blue font). Point-by-point responses to the referee are listed below. Besides that, we attach an additional manuscript and Supplementary material in which the major text changes are marked in red. We believe that the revised manuscript has taken full account of the points raised by the reviewer, and we hope you will be satisfied with the revised version of this paper.

Best regards,

The authors

REVIEWER COMMENTS

Reviewer #1 (Remarks to the Author):

1. *I want to thank authors for the answers. The authors clarify that Fabry-Perot mode and Tamm optical state should form new hybrid mode what they called hybrid Tamm cavity. In the Fig.1 (b) and (h) showed resonant modes which according to authors corresponds for the hybrid Tamm-Cavity mode, however, **at the substrate thickness $d=510$ um in both pictures (b) and (h) from the optical response I do not see any evidence of hybridization.** Meanwhile, for about 440 um thickness (0.35-0.4 THz range) a **hybridization can be clearly seen and anti-crossing effect of some cavity mode (as I guess) with photonic band edge of the distributed Bragg reflector.** If authors claim that this is hybrid mode, then in what kind of coupling regime this mode is? For me the evidence of hybridization of Tamm state and cavity mode is still unclear.*

Reply: We agree with the reviewer's comments. Sincere thanks to the editor and Reviewer 1, for their help, which has allowed us to deeply understand the meaning of hybridization of Tamm state and Tamm-cavity modes. For the “Hybrid” in our manuscript, we are referring to the structural hybrid, not the mode-coupling hybridization. In physics, the structure proposed in this manuscript can be seen as a hybrid Tamm cavity formed by inserting a dielectric layer into a pure Tamm cavity, and the terahertz detector is prepared on this dielectric layer. The design relies on the position of the detector from the cavity side of this Tamm-cavity hybrid structure. The Tamm mode of the hybrid structure results in a sharper reflection dip than achievable using a pure Tamm cavity structure (without cavity). At the resonance wavelength of the Tamm mode in the DBR stop band region, the Au film and top DBR trap light effectively in the cavity, resulting in local enhancement of electric field at the detector. The electric field intensity is significantly enhanced in the hybrid Tamm cavity and a high Q-factor is achieved. In addition, the resonance wavelength of the hybrid structure can be shifted along the DBR stop band simply by varying the cavity thickness, making a wide range of detection wavelengths easily obtainable. The outstanding optical properties and highly sensitive detection of this easy-to-fabricate hybrid structure makes it a promising candidate for use in a variety of THz devices applications.

In fact, Reviewer 1 provided us with many insights in the first round of review comments. Unfortunately, we did not fully understand the reviewer's suggestions. Therefore, we once again thank Reviewer 1. In order to dispel the confusion of Reviewer 1, we have updated our concepts and highlight three points according to the reviewer's comments:

1. We apologize for any confusion caused to Reviewer 1 due to the lack of rigor in our previous responses. It must be pointed out that the hybrid structure, rather than the coupling hybridization of cavity modes, is proposed in this manuscript as a Tamm-cavity mode. Specifically, the detector on a silicon substrate is inserted into the Tamm structure to form the MS (LH)³ structure (M is the metal layer, L is the air layer, H is the silicon layer), where the S layer is the thickness of the silicon spacer adjacent to the metal, which is also the substrate of the detector. There is no doubt that the reflection dips generated in the bandgap is caused by Tamm state mode [1-

12]. Within the bandgap of DBR, only Tamm mode exists, so there is no claim of coupling between Tamm mode and FP mode; However, as mentioned in Ref. 56: “The frequency of the Tamm modes is determined by phase matching of a round trip inside the cavity, similarly to common Fabry Perot activities”, as indicated by the white dashed line in **Fig. 1 (h)**.

2. **Fig. 1** in the manuscript shows the calculated results for the Tamm-cavity terahertz detector structure. In the electromagnetic field simulation, the influence of the sensor film (100 nm Nb₅N₆) does not need to be considered at all. Because the thickness of the Nb₅N₆ film for the detector is around 100 nm, its influence on the electric field distribution can be ignored compared to the calculated terahertz wavelength. At the same time, Tamm plasmonic cannot be excited in the terahertz band, so the resonant mode we constructed should be Tamm mode [4-7]. The detector is inserted into the Tamm structure, while requiring the maximum electric field at the detector to maintain its maximum optical response. As pointed out by the reviewer 1, a hybridization and anti-crossing effect occurs outside the DBR stop band. As will be analyzed later, this splitting is caused by the coupling between the leaky Tamm cavity mode and the detector's substrate cavity (FP mode), and no electric field stronger than Tamm cavity mode in the DBR stop band. However, hybridization coupling mode outside the DBR stop band is not the focus of this study for the poor electric field and low optical response of the detector correspondingly. Thanks to the Reviewer 1, we have noticed that the hybridization and anti-crossing effect have received extensively attention from researchers [12-25]. We express our sincere gratitude to the reviewer, for he/she has helped us understand the true meaning of mode hybridization, anti-crossing, and strong coupling.
3. In the terahertz range, the integration preparation of this high refractive index contrast DBR cavity with a detector is extremely challenging. This work successfully achieved the integration of a THz detector with the Tamm cavity. By utilizing the high Q and tunable resonance characteristics of Tamm cavity [2,3,6,9-12], a THz detector with high sensitivity, high resonance Q value, and adjustable frequency was achieved. As point out by the Reviewer 1, Tamm cavities have been widely studied in the past decade in the visible and infrared range [1-3,8-25]. They are composed of a metal layer covering a distributed Bragg reflector (DBR) made of a stack of high and low refractive index quarter-wavelength layers. The Tamm electromagnetic mode arises at the interface between the metal layer and the periodic stack and is due to the metal inducing a break of periodicity in the structure. In the infrared range, the DBR is typically grown by molecular beam epitaxy or PECVD, such as GaAs/Al_{0.95}Ga_{0.05}As [13]. These fabrication methods are not directly scalable to the THz range, since the large thickness of quarter-wavelength layers at terahertz frequencies (~20 μm) is not compatible with epitaxial or PECVD techniques, and THz Tamm cavity schemes have only been proposed theoretically. Furthermore, it is very difficult to integrate and prepare DBR cavities with high refractive index contrast media and the process compatibility of detector integration also needs to be considered, which is undoubtedly a major obstacle. The focus of this paper is on the integrated Tamm cavity detector, which utilizes local enhancement of the Tamm cavity to improve the optical response

of the detector. Furthermore, by changing the substrate thickness of the detector, a detector with adjustable resonant frequency is achieved. These research results have not been reported yet.

2. *Meanwhile, for about 440 μm thickness (0.35-0.4 THz range) a hybridization can be clearly seen and anti-crossing effect of some cavity mode (as I guess) with photonic band edge of the distributed Bragg reflector. If authors claim that this is hybrid mode, then in what kind of coupling regime this mode is? For me the evidence of hybridization of Tamm state and cavity mode is still unclear.*

Reply: Greats thanks for the comments. We agree with the reviewer's comments. The reflection spectrum was also calculated. In order to further clarify and understand the characteristics of these resonant Tamm modes, the reflection map and spatial electromagnetic field distribution of this hybrid Tamm cavity ($d_{\text{substrate}} = 510 \mu\text{m}$, see **Fig. R1**) and pure Tamm cavity ($d_{\text{substrate}} = \frac{\lambda}{4n_{\text{Si}}} = 33 \mu\text{m}$, see **Fig. R2**) are also calculated. **Fig. R1(b)** show the reflection spectrum of the hybrid Tamm cavity when $d_{\text{substrate}} = 510 \mu\text{m}$, $450 \mu\text{m}$ and $33 \mu\text{m}$ respectively. **The leaky Tamm modes has low quality factor Q and large reflectivity due to the imperfect reflection outside the DBR stopband [1-2].** As shown in the inset of **Fig. R1(b)**, within the DBR bandgap, the electric field of the hybrid cavity (at 0.47 THz) is much larger than that of the pure Tamm cavity (at 0.65 THz), almost six times larger, which confirms the highly localized enhancement and high Q value of the electric field in the hybrid Tamm cavity. Meanwhile, the existence of multiple Tamm states within the DBR bandgap is demonstrated in hybrid Tamm cavity for $d_{\text{substrate}} = 510 \mu\text{m}$, $450 \mu\text{m}$. However, in the pure Tamm cavity as demonstrated in **Fig. R2**, there is only one Tamm state within the DBR bandgap and multiple leaky Tamm modes exist outside the DBR bandgap. It is found that a Tamm states is periodic resonance with the variation of the top-layer thickness of the silicon dielectric for a given wavelength for both pure and hybrid cavity structure, which is precisely determined by Formula 2 in the main text of the manuscript.

As shown in **Fig. R1(c)**, multiple splits occur at certain frequencies. This phenomenon is caused by the coupling of various **leaky Tamm modes** outside of the DBR stopband (**Fig. R1(e)**, red dashed box in **Fig. R1(c)** and left red arrow in **Fig. R2(c)**) and **substrate cavity modes** (FP modes) (**Fig. R1(f)**). The split and hybridization of these two modes in 0.2-1.2 THz are shown in **Fig. R1(c)**. One of these at around 0.35 THz is enlarged and depicted in **Fig. R1(d)** and anti-crossing effect can be clearly seen, indicating a strong coupling. **To explain the reason for the hybrid mode generated outside the band gap of DBR, we split the hybrid Tamm cavity into two parts, one of which is the pure Tamm cavity ($d_1 = \frac{\lambda}{4n_{\text{Si}}} = 33 \mu\text{m}$), the other part is the silicon FP cavity ($d_2 = d_{\text{substrate}} - \frac{\lambda}{4n_{\text{Si}}}$), and the reflection spectra of these three cavities were calculated separately.**

As can be seen from **Fig. R1(g)**, it is precisely due to the mode coupling of these two cavities that splitting and anti-crossing effect occurs, which confirms the reviewer's point of a hybridization for about 440 μm thickness (0.35-0.4 THz range).

Fig. R1 (a) The scheme of the hybrid Tamm cavity with a thickness tunable detector substrate embedded in a Tamm structure. (b) The reflection spectrum of the hybrid structure when $d_{\text{substrate}} = 510 \mu\text{m}$, $450 \mu\text{m}$ and $33 \mu\text{m}$. The inset is the calculated spatial electromagnetic field distribution of this hybrid Tamm structure ($d_{\text{substrate}} = 510 \mu\text{m}$) and pure Tamm cavity ($d_{\text{substrate}} = \frac{\lambda}{4n_{\text{Si}}} = 33 \mu\text{m}$). (c) The relation between the reflection and the thickness of detector substrate at 0.15 THz - 1.2 THz. Multiple anti-crossing splits occur outside the DBR stopband. (d) Enlarged image of an anti-crossing split in the red box in (c). (e) Calculated reflection of leaky Tamm mode in a pure Tamm cavity. (f) Calculated reflection of FP cavity mode in FP cavity. Considering the detector substrate is the combination of a typical DBR layer and FP cavity. The thickness of the FP cavity is $d_{\text{substrate}} - \lambda/4n_{\text{Si}}$. (g) Calculated reflection of a coupled leaky Tamm and FP cavity mode with anti-crossing effect in a hybrid Tamm cavity.

Fig. R2 (a) The scheme of a pure Tamm cavity with a 3-pair DBR. (b) Reflection spectrum of (a). The insets are the electrical distribution of leaky Tamm mode at 0.36 THz and Tamm mode at 0.65 THz. (c) Relation between reflection and $d_{substrate}$ of Fig.R1 (a), wherein the anti-crossing splits occurs when the FP cavity modes resonate with the leaky Tamm modes.

In order to further clarify and understand the characteristics of these resonant Tamm modes and follow the reviewer's suggestions, the analysis and discussion on the Tamm modes and its coupling with the detector's substrate cavity were added as Supplementary Note S3 to the supplementary material. Once again, we sincerely thank the reviewer's guidance and suggestions.

In addition, it is interesting that the FP cavity modes resonate with the leaky Tamm mode rather than the Tamm mode as depicted in Fig. R1 and Fig. R2. As an additional extension, under the guidance and inspiration of Reviewer 1, in order to further understand the hybridization mechanism and apply the hybridization of resonant cavity modes, we calculated and analyzed the phenomenon of mode hybridization caused by multi cavity coupling. **Fig. R3 (a)** represents a new structure which consists of the hybrid structure in the manuscript, a cavity and an DBR added on top. The added DBR and cavity, together with DBR in the hybrid structure, forms a FP cavity to excite FP cavity modes within the cavity. Meanwhile, the multiple optical Tamm modes are still excited in the 510 μm detector substrate. Then multiple mode hybridizations are observed with an anti-crossing splitting within the DBR stopband, as depicted in **Fig. R3 (b)**. One of these at around 0.65 THz is enlarged and depicted in **Fig. R3 (d)** and anti-crossing effect can be clearly seen, indicating a strong coupling.

Fig. R3 (a) The scheme of the hybrid Tamm cavity with a cavity and a DBR added on top. (b) Relation between reflection and the thickness of the cavity (d_{cavity}). The anti-crossing splits occur when the Tamm mode in the hybrid Tamm cavity resonates with the cavity mode excited in the added cavity. (c) Reflection spectrum of (a) when $d_{\text{cavity}} = 475 \mu\text{m}$. (d) Enlarged image of an anti-crossing split in the white box in (b).

Notably, the coupling of optical modes and the coupling of photon and quasiparticle is similar but slightly different. The photon-quasiparticle coupling in [13-19], which requires photon to excite exciton in WS_2 monolayer then couple with it, meaning that the optical mode excited should localize the photon at where the monolayer is. However, the coupling of optical modes requires two modes excited in on optical cavity, at the very least, in our structure, the modes can only be excited at different location. The hybrid Tamm cavity in this work is intended to achieve high electrical intensity at interface where the THz detector is deposited, boosting its optical response. Our structure can also benefit many applications such as emitters [20-21,24], absorber [23] and surface enhanced resonance Raman spectroscopy [22]. It is indeed an interesting phenomenon and we would like to see how the 2D device made of 2D material like TMDCs embedded in our hybrid Tamm cavity performs in THz range [25], since the bandgap of TMDCs can be suppressed to THz band. We will study it in our future work.

As pointed out by the reviewer, we use “Tamm mode” to denote the modes excited within our structure. Based on the suggestions of the reviewers and to enhance the readability of the article, we also revise the explanation about the splinting in the seventh paragraph in the

section of Device design, and it reads:

“The splitting observed within the 0.35 – 0.4 THz region has anti-crossing effect in Fig. 1(h), indicating a hybridization mode which is the strong coupling of FP cavity mode excited in detector substrate and leaky Tamm mode excited in a pure Tamm structure. As analyzed in Supplementary Note S3, this splitting is caused by the coupling between the leaky Tamm cavity mode and the detector's substrate cavity (FP mode). The leaky Tamm modes has low quality factor Q and large reflectivity due to the imperfect reflection outside the DBR stopband [56, 73], localizes its energy within the DBR structure, giving space for the detector substrate to excite FP cavity modes. Consequently, the hybrid mode exhibits leakiness, leading to lower electrical intensity.”

Reference

- S1. Kavokin, A. V., Baumberg, J. J., Malpuech, G. & Laussy, F. P. *Microcavities* Oxford Univ. Press (2007).
- S2. R. Bruckner, M. Sudzius, S.I. Hintschich, H. Fröb, V.G. Lyssenko, K. Leo, Hybrid optical Tamm states in a planar dielectric microcavity, *Phys. Rev. B* 83, 033405 (2011).
- S3. Zhiyu Wang, J. Kenji Clark, Ya-Lun Ho, Bertrand Vilquin, Hirofumi Daiguji, and Jean-Jacques Delaunay, Narrowband Thermal Emission Realized through the Coupling of Cavity and Tamm Plasmon Resonances, *ACS Photonics*, 5, 2446-2452 (2018).
- S4. Simon Messelot, Solen Coeymans, Jérôme Tignon, Sukhdeep Dhillon, and Juliette Mangeney, High Q and sub-wavelength THz electric field confinement in ultrastrongly coupled THz resonators, *Photon. Res.* 11, 1203-1216 (2023)
- S5. Simon Messelot, Clémentine Symonds, Joël Bellessa, Jérôme Tignon, Sukhdeep Dhillon, Jean-Blaise Brubach, Pascale Roy, and Juliette Mangeney, Tamm Cavity in the THz Spectral Range, *ACS Photonics* 7 (10), 2906-2914 (2020).
- S6. Symonds, C., Azzini, S., Lheureux, G. et al. High quality factor confined Tamm modes. *Sci Rep* 7, 3859 (2017). <https://doi.org/10.1038/s41598-017-04227-1>
- S7. Gazzano O, de Vasconcellos S M, Gauthron K, Symonds C, Bloch J, Voisin P, Bellessa J, Lemaître A and Senellart P, *Phys. Rev. Lett.* 107, 247402 (2011)
- S8. K. Leosson, S. Shayestehaminzadeh, T.K. Tryggvason, A. Kosoy, B. Agnarsson, F. Magnus, S. Olafsson, J.T. Gudmundsson, E.B. Magnusson, I.A. Shelykh, Comparing resonant photon tunneling via cavity modes and Tamm plasmon polariton modes in metal-coated Bragg mirrors, *Opt. Lett.* 37, 4026 (2012).
- S9. X.-L. Zhang, J.-F. Song, J. Feng, H.-B. Sun, Spectral engineering by flexible tunings of optical Tamm states and Fabry–Perot cavity resonance, *Opt. Lett.* 38, 214382 (2013).
- S10. H. Zhou, G. Yang, K. Wang, H. Long, P. Lu, Multiple optical Tamm states at a metal dielectric mirror interface, *Opt. Lett.* 35, 4112 (2010).
- S11. Ye Ming Qing, Hui Feng Ma, and Tie Jun Cui, Flexible control of light trapping and localization in a hybrid Tamm plasmonic system, *Opt. Lett.* 44, 3302-3305 (2019)
- S12. Rajesh V. Nair, The interaction between optical Tamm state and microcavity mode in a planar

- hybrid plasmonic-photonic structure, *Photonics and Nanostructures - Fundamentals and Applications*, 36,100702 (2019). <https://doi.org/10.1016/j.photonics.2019.100702>
- S13. Weisbuch, C., Nishioka, M., Ishikawa, A. & Arakawa, Y. Observation of the coupled exciton-photon mode splitting in a semiconductor quantum microcavity. *Phys. Rev. Lett.* 69, 3314–3317 (1992).
- S14. S.K. Shaid-Ur Rahman, T. Klein, S. Klemmt, Jürgen Gutowski, D. Hommel, K. Sebald, Observation of a hybrid state of Tamm plasmons and microcavity exciton polaritons, *Sci. Rep.* 6 (2016) 34392.
- S15. Lackner, L., Dusel, M., Egorov, O.A. et al. Tunable exciton-polaritons emerging from WS₂ monolayer excitons in a photonic lattice at room temperature. *Nat Commun* 12, 4933 (2021).
- S16. Liu, X., Galfsky, T., Sun, Z. et al. Strong light–matter coupling in two-dimensional atomic crystals. *Nature Photon* 9, 30–34 (2015). <https://doi.org/10.1038/nphoton.2014.304>
- S17. Waldherr, M., Lundt, N., Klaas, M. et al. Observation of bosonic condensation in a hybrid monolayer MoSe₂-GaAs microcavity. *Nat Commun* 9, 3286 (2018).
- S18. Schneider, C., Glazov, M.M., Korn, T. et al. Two-dimensional semiconductors in the regime of strong light-matter coupling. *Nat Commun* 9, 2695 (2018).
- S19. Lundt, N., Klemmt, S., Cherotchenko, E. et al. Room-temperature Tamm-plasmon exciton-polaritons with a WSe₂ monolayer. *Nat Commun* 7, 13328 (2016).
- S20. Leuthold, J., Dorodnyy, A. On-demand emission from Tamm plasmons. *Nat. Mater.* 20, 1595–1596 (2021).
- S21. He, M., Nolen, J.R., Nordlander, J. et al. Deterministic inverse design of Tamm plasmon thermal emitters with multi-resonant control. *Nat. Mater.* 20, 1663–1669 (2021).
- S22. Sreekanth, K.V., Perumal, J., Dinish, U.S. et al. Tunable Tamm plasmon cavity as a scalable biosensing platform for surface enhanced resonance Raman spectroscopy. *Nat Commun* 14, 7085 (2023).
- S23. M. He, J. R. Nolen, J. Nordlander, A. Cleri, G. Lu, T. Arnaud, N. S. McIlwaine, K. Diaz-Granados, E. Janzen, T. G. Folland, J. H. Edgar, J.-P. Maria, J. D. Caldwell, Coupled Tamm Phonon and Plasmon Polaritons for Designer Planar Multiresonance Absorbers. *Adv. Mater.*, 35, 2209909 (2023). <https://doi.org/10.1002/adma.202209909>
- S24. He, M., Nolen, J.R., Nordlander, J. et al. Deterministic inverse design of Tamm plasmon thermal emitters with multi-resonant control. *Nat. Mater.* 20, 1663–1669 (2021). <https://doi.org/10.1038/s41563-021-01094-0>
- S25. Huang, W., Folland, T.G., Sun, F. et al. In-plane hyperbolic polariton tuners in terahertz and long-wave infrared regimes. *Nat Commun* 14, 2716 (2023). <https://doi.org/10.1038/s41467-023-38214-0>

Finally, sincere thanks again to the reviewer for the time spent on our manuscript and providing us with many useful suggestions. We hope that in the revised manuscript, we have addressed the reviewer’s concern in a manner that is somewhat satisfactory.

Reviewer #3 (Remarks to the Author):

Authors carefully addressed the points raised in the previous review iteration. I therefore thank the authors and recommend the article for publication.

Reply: We are very grateful to the referee for the positive comments on our revised version. It recognizes the efforts we have put into amending the manuscript. We thank the reviewer again for valuable comments that improved the scientific quality of this paper.

REVIEWERS' COMMENTS

Reviewer #1 (Remarks to the Author):

The authors improve the manuscript and their understanding about mode coupling in such structures. So, finally, the reader can find out that Tamm states with cavity modes support structures was employed to enhance detection of the signal. However, the application of optical Tamm states to the sensing enhancement as well as their coupling with other modes was already demonstrated in VIS-IR region.

REVIEWER COMMENTS

Reviewer #1 (Remarks to the Author):

The authors improve the manuscript and their understanding about mode coupling in such structures. So, finally, the reader can find out that Tamm states with cavity modes support structures was employed to enhance detection of the signal. However, the application of optical Tamm states to the sensing enhancement as well as their coupling with other modes was already demonstrated in VIS-IR region.

Reply: Sincere thanks to the editor and Reviewer 1, for their help, which has allowed us to deeply understand the meaning of hybridization of Tamm state and Tamm-cavity modes.

As point out by the Reviewer 1, Tamm cavities have been widely studied in the past decade in the visible and infrared range [1-7]. They are composed of a metal layer covering a distributed Bragg reflector (DBR) made of a stack of high and low refractive index quarter-wavelength layers. The Tamm electromagnetic mode arises at the interface between the metal layer and the periodic stack and is due to the metal inducing a break of periodicity in the structure. In the infrared range, the DBR is typically grown by molecular beam epitaxy or PECVD, such as GaAs/Al_{0.95}Ga_{0.05}As [2]. These fabrication methods are not directly scalable to the THz range, since the large thickness of quarter-wavelength layers at terahertz frequencies (~20 μm) is not compatible with epitaxial or PECVD techniques, and THz Tamm cavity schemes have only been proposed theoretically. **Furthermore, it is very difficult to integrate and prepare DBR cavities with high refractive index contrast media and the process compatibility of detector integration also needs to be considered, which is undoubtedly a major obstacle.** The focus of this paper is on the integrated Tamm cavity with THz detector. Furthermore, by changing the substrate thickness of the detector, a detector with adjustable resonant frequency is achieved. In fact, **We have fully elaborated on these contents in the introduction section of the manuscripts.**

Reference

1. Kavokin, A. V., Baumberg, J. J., Malpuech, G. & Laussy, F. P. *Microcavities* Oxford Univ. Press (2007).
2. Weisbuch, C., Nishioka, M., Ishikawa, A. & Arakawa, Y. Observation of the coupled exciton-photon mode splitting in a semiconductor quantum microcavity. *Phys. Rev. Lett.* **69**, 3314–3317 (1992).
3. Schneider, C., Glazov, M.M., Korn, T. et al. Two-dimensional semiconductors in the regime of strong light-matter coupling. *Nat Commun* **9**, 2695 (2018).
4. Leuthold, J., Dorodnyy, A. On-demand emission from Tamm plasmons. *Nat. Mater.* **20**, 1595–1596 (2021).
5. He, M., Nolen, J.R., Nordlander, J. et al. Deterministic inverse design of Tamm plasmon thermal emitters with multi-resonant control. *Nat. Mater.* **20**, 1663–1669 (2021).
6. Sreekanth, K.V., Perumal, J., Dinish, U.S. et al. Tunable Tamm plasmon cavity as a scalable

biosensing platform for surface enhanced resonance Raman spectroscopy. *Nat Commun* **14**, 7085 (2023).

7. Huang, W., Folland, T.G., Sun, F. et al. In-plane hyperbolic polariton tuners in terahertz and long-wave infrared regimes. *Nat Commun* **14**, 2716 (2023).

Finally, sincere thanks again to the reviewer for the time spent on our manuscript and providing us with many useful suggestions. We hope that we have addressed the reviewer's concern in a manner that is somewhat satisfactory.